# Functional contrast across the gray-white matter boundary

Muwei Li ®[1,2] ✉, Lyuan Xu[1,3], Soyoung Choi[1,2], Yuanyuan Qin[4], Fei Gao[5], Kurt G. Schilling ®[1,2], Yurui Gao ®[1,6], Zhongliang Zu ®[1,2], Adam W. Anderson[1,2,6], Zhaohua Ding ®[1,3,6,7] & John C. Gore[1,2,6]

Functional magnetic resonance imaging studies have traditionally focused on gray matter, overlooking white matter despite growing evidence that functional blood oxygenation-level dependent effects also occur there. In particular, functional coupling across the gray-white matter boundary, an interface between local and global processing, remains poorly understood. This study introduces two metrics: gray-white matter functional connectivity, which captures temporal synchrony across the boundary, and gray-white blood oxygenation-level dependent power ratio, which reflects differences in signal amplitude. Gray-white matter functional connectivity aligns with patterns of myelination, long-range connectivity, and sensorimotor organization, suggesting efficient signal transmission. In contrast, the power ratio shows an inverse pattern, with higher values in higher-order regions, possibly reflecting increased metabolic demands in white matter. It also increases with age (8 to 21 years), suggesting developmental shifts in energetic demands. Together, these metrics highlight distinct yet complementary roles of signal fidelity and energy modulation at the gray-white matter boundary.

The human brain is intricately organized into gray matter (GM) and white matter (WM), two structurally and functionally distinct tissues that both play essential roles in neural processing. Gray matter consists predominantly of neuronal cell bodies, dendrites, unmyelinated axons, and synapses, which are responsible for local information processing and integration, while white matter comprises myelinated axons that enable long-range signal transmission between distant GM regions. The boundary between GM and WM, linking local cortical processing with broader white matter pathways that support large-scale brain communication, represents a unique and critical interface for understanding brain connectivity. Structural studies have examined the GM-WM boundary using T1-weighted (T1w) imaging. These studies primarily focus on the structural contrast between GM and WM, measured as the ratio of T1w intensity between adjacent GM and WM points aligned perpendicular to the boundary[1]. This contrast has been found to vary with myelination[2] and water content[3] in WM, and it can become blurred with aging[1], Alzheimer's disease[3], and Parkinson's disease[4]. In addition, due to its intricate microstructural organization, the boundary is particularly vulnerable to various diseases and injuries, such as abscesses[5], metastasis[6], and diffusive axonal injury[7]. Despite its importance, the nature of functional coupling across the boundary, which potentially reflects alterations in BOLD signals as they traverse from one type of tissue to another, remains poorly understood. Given the distinct metabolic, neurovascular, and structural properties of GM and WM, it is expected that the characteristics of functional MRI (fMRI) signals, based on the BOLD (blood oxygenation-level dependent) effect, will differ on either side of the boundary.

[1]Vanderbilt University Institute of Imaging Science, Vanderbilt University Medical Center, Nashville, TN, USA. [2]Department of Radiology and Radiological Sciences, Vanderbilt University Medical Center, Nashville, TN, USA. [3]Department of Electrical and Computer Engineering, Vanderbilt University, Nashville, TN, USA. [4]Department of Radiology, Tongji Hospital, Tongji Medical College, Huazhong University of Science and Technology, Wuhan, China. [5]Department of Radiology, Shandong Provincial Hospital Affiliated to Shandong First Medical University, Jinan, China. [6]Department of Biomedical Engineering, Vanderbilt University, Nashville, TN, USA. [7]Department of Computer Science, Vanderbilt University, Nashville, TN, USA. ✉e-mail: m.li@vumc.org

Understanding such differences is key to revealing how the brain integrates local processing (in GM) and global information transmission (through WM), and how these processes might vary across regions of the brain. Unfortunately, such insights have been limited to only one side of the boundary, as fMRI studies have historically focused almost exclusively on the GM. This focus stems from our observations that BOLD signals, which reflect changes in blood flow, volume, and oxygenation, are more pronounced in GM due to its higher vascularization and metabolic activity[8], thus leading to BOLD effects in WM being overlooked. However, recent studies have challenged this notion. Clear evidence shows that signals representing neural activities within WM are measurable using appropriate analytical methods[9–18], and resting-state BOLD synchronization between different WM regions reveals interacting networks and communities[19–22]. These findings provide opportunities for further exploration of functional interactions between the two sides of the boundary. Such interactions may reflect brain adaptations to support efficient signal propagation, influenced by factors such as myelination, regional functional load, metabolism, and neurochemical activity.

Here we introduce two metrics of functional activity that complement traditional structural contrast measures, and which quantify the similarities and differences in BOLD signals between two GM and WM points (voxels) that are near, and aligned perpendicular, to the GM-WM boundary, thereby enabling characterization of functional contrast across this critical interface. These features are characterized in two ways. First, we introduce gray-white matter functional connectivity (GWFC) by calculating the Pearson correlation between the BOLD signals of those two points, which we propose reflects the fidelity of signal transmission across the boundary. High values of GWFC reflect strong temporal similarity between BOLD signals in neighboring GM and WM regions. Second, we define the gray-white BOLD power ratio (GWBPR) as the ratio of the fractional amplitudes of low-frequency fluctuations[23] (fALFF) between these two points, to capture how the spectral power of the BOLD signal alters across the boundary. These two measurements may reflect distinct biological processes. GWFC reflects signal synchronization and likely relates to local infrastructure that maintains the accuracy of signal transmission, such that both GM and WM elicit correlated vascular responses. In contrast, the GWBPR reflects changes in signal magnitudes, which may be more associated with the different magnitudes of oxygenation and blood volume changes required to support the metabolic energy demands of signal transmission in WM and GM[24]. Together, these metrics provide complementary insights into the efficiency and integrity of signal propagation across the GM-WM boundary. To further contextualize the functional patterns we observe, we also examined the regional homogeneity[25] (ReHo), which captures the functional connectivity among neighboring GM points that are aligned tangentially to the boundary. By comparing GWFC to ReHo, we aim to demonstrate that GWFC provides unique insights into functional coupling across the GM-WM boundary, offering information that cannot be captured by connectivity measurements confined to GM.

Our results reveal a significant correlation between GWFC and myelin content, with higher GWFC observed in regions with greater myelin content (T1-weighted to T2-weighted ratio). In contrast, ReHo, which is primarily driven by local intra-GM connectivity, shows a weaker correlation with myelin content, indicating BOLD contributions to ReHo reflect a different type of neural interaction. Additionally, long-range functional connectivity (the mean FC between a GM point and all WM points across the brain) is better predicted by GWFC than by ReHo. GWFC was also found to be positively correlated with regions involved in sensorimotor functions, as defined by a previously reported sensorimotor-association axis map[26]. It is also positively correlated with the density of neurotransmitter receptors associated with specific neurochemical processes (e.g., 5-HT1F, β1, H2) and negatively correlated with transmodal regions and receptors linked to higher-order cognition (e.g., 5-HT3A, 1A, 7).

GWBPR exhibits an approximately inverse distribution compared to GWFC, showing negative correlations with myelin content but positive correlations with higher-order functions. Building on these findings, we further examined how GWBPR varies with age during late childhood and adolescence (8–21), a developmental period during which lower-order functions are largely developed, while higher-order functions are still maturing[27]. We observed widespread regions showing a positive correlation between GWBPR and age, with most of these regions being associated with higher-order functions.

In conclusion, the functional contrasts proposed here offer a unique perspective on brain connectivity by capturing how functional signals are transmitted across the GM-WM boundary. Their distributions are related to structural (myelination), functional (sensorimotor vs. association regions), and neurochemical (receptor distribution) tissue characteristics, and they provide a more comprehensive view of specific aspects of brain connectivity that may have both clinical and theoretical significance.

## Results

Two functional contrasts, GWFC and GWBPR, were computed for each vertex on the GM-WM boundary. These metrics were derived from resting-state fMRI data from two publicly available cohorts: the Human Connectome Project–Development[28] (HCP-D; $n = 571$, ages 8–21) and the Human Connectome Project-Young Adult[29] (HCP-Y; $n = 687$, ages 22–35). As shown in Fig. 1, GWFC measures the Pearson correlation between BOLD signals at corresponding GM and WM points along a perpendicular axis, while GWBPR quantifies the ratio of fALFF between these points. Additionally, ReHo (Regional Homogeneity) was calculated to assess functional connectivity along the tangential direction within GM. It should be noted that the WM points were sampled 1 mm beneath the GM-WM boundary, while the fMRI data have a 2 mm isotropic resolution. Although trilinear interpolation was used to estimate sub-voxel BOLD values, some partial volume effects are possible, especially near subcortical structures. We addressed this by zeroing out intensities from subcortical voxels, though residual signal contamination cannot be entirely excluded.

### Distribution of GWFC and its relationship to myelin content

The spatial distribution of GWFC averaged across all subjects is shown in Fig. 2. The highest correlations are observed in sensory areas, including the primary visual and primary sensory cortex, while the lowest intensities are seen in the anterior cingulate cortex. In contrast, ReHo exhibits more consistently high intensities across the brain, though noticeably low intensities are found in the hippocampus, inferior parietal cortex, and anterior TE2 area. However, similar to GWFC, ReHo also shows noticeably high intensity in the primary sensory cortex. To further contextualize these patterns, we compared GWFC and ReHo with cortical myelin content, estimated using the T1w/T2w ratio from the HCP-Y structural dataset. Visual inspection reveals that the distribution of GWFC is highly consistent with that of myelin content. Correlation analyses confirm that the myelin map is more strongly associated with GWFC ($r(358) = 0.40$, $p = 2.7470\mathrm{e}{-15}$, 95% CI = [0.31, 0.48]) than with ReHo ($r(358) = 0.32$, $p = 2.9541\mathrm{e}{-10}$, 95% CI = [0.23, 0.41]). To further assess the robustness and reproducibility of this pattern, we conducted a validation analysis. Specifically, we randomly selected 100 subjects from the full dataset ($n = 687$) and repeated the group-level GWFC, ReHo, and myelin computations across the cortical surface. We then computed the correlation of GWFC/ReHo with the myelin map across regions. This process was repeated 30 times. As shown in Figure S1, across these 30 random subsets, GWFC consistently showed stronger correlations with myelin than ReHo, suggesting that the observed trend is stable and reproducible.

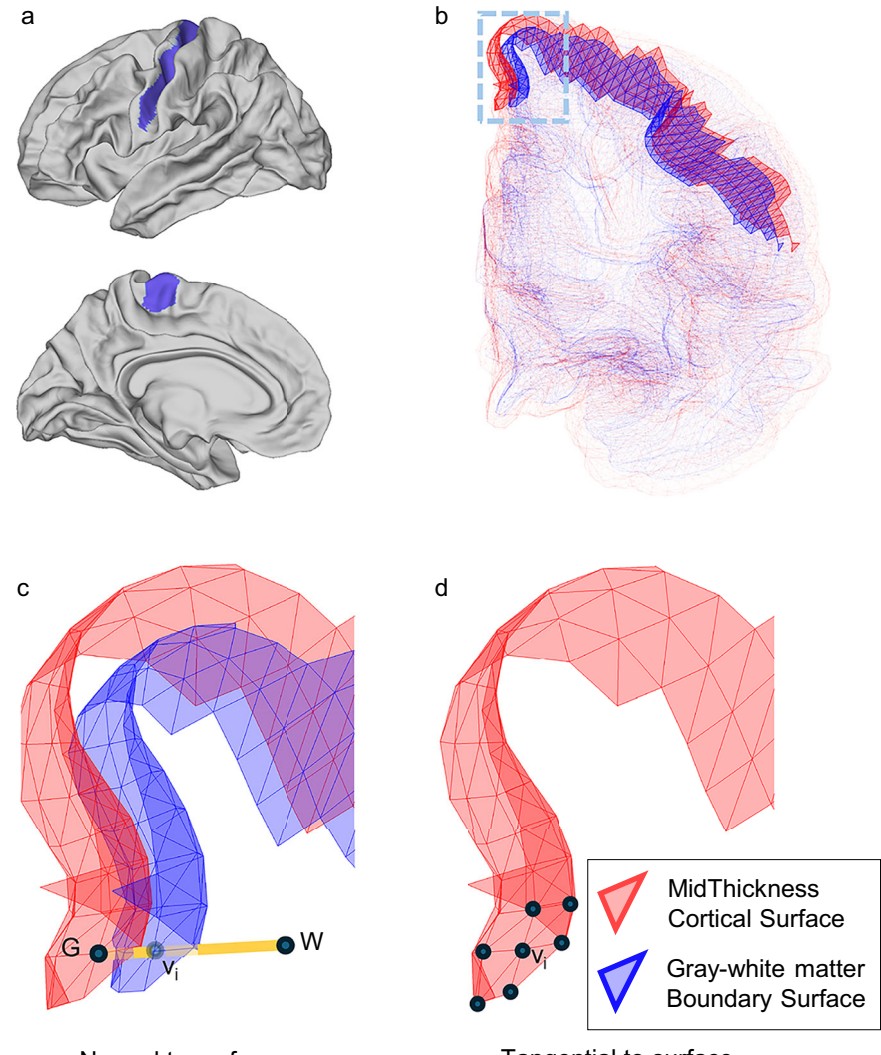

**Fig. 1 | Overview of functional contrast across the gray-white matter boundary.**
**a** The white matter surfaces are shown from a lateral view, highlighting the primary
motor cortex in blue. **b** Mesh surface of mid-thickness cortex (red) and gray-white
matter boundary (blue). **c** Schematic of functional contrast measures normal to the
surface. For a given vertex $v_i$ on the boundary surface, its corresponding GM vertex
(G) and WM point (W) are identified along the line perpendicular to the surface.
**d** Schematic of functional connectivity measures tangential to the surface. ReHo is
calculated for each vertex $v_i$ on the mid-thickness surface by measuring the cor-
relation of BOLD signals among neighboring vertices, which are connected along
the tangential direction to the surface. ReHo regional homogeneity.

It is important to note that myelin content was not measured at a
single vertex; instead, for each vertex, a ribbon was reconstructed,
covering the entire space between the pial and white matter surfaces.
To ensure that this result is not driven by the method of estimating
myelin using values averaged across cortical layers, we repeated the
analysis using myelin maps computed at the mid-thickness surface
only. This test was conducted on a subset of 100 HCP-Y subjects (the
first 100 sorted by study ID). As shown in Figure S2, the resulting
correlation between mid-thickness myelin and the full-layer myelin
map was high (r(358) = 0.9787, p = 3.3179e-248, 95% CI = [0.97, 0.98]),
confirming that our findings are robust to how myelin is sampled
across the cortex.

### Relationship between GWFC and long-range FC
To assess the relationship between local and global functional organi-
zation, we introduced a long-range FC measure for each GM vertex.
This was computed as the mean Pearson correlation between the BOLD
signal of that vertex and all WM voxels across the brain, capturing
the extent of global integration through WM pathways. As shown in
Fig. 3, although both GWFC and ReHo show a noticeable correlation

with long-range FC, the correlation is stronger for GWFC (r(358) = 0.67,
p = 7.5282e-49, 95% CI = [0.61, 0.73]), likely due to its ability to capture
connections extending into WM, reflecting the direct role of this
interface in facilitating communication across distant brain regions. In
contrast, the correlation between ReHo and long-range FC is weaker
(r(358) = 0.42, p = 3.7826e-17, 95% CI = [0.34, 0.51]), presenting weaker
evidence of local synchronization within GM playing a role in long-range
connections. There is a potential bias in the comparison between ReHo
and GWFC due to the difference in calculation methods. ReHo is based
on KCC, whereas GWFC and long-range FC are computed using Pear-
son's correlation. To address this, we tested an alternative version of
ReHo (based on 100 subjects) by replacing KCC with the mean of
pairwise Pearson correlations among neighboring voxels. As shown in
Figure S3, the two ReHo measures are highly correlated
(r(358) = 0.9991, p < 2.2e−16, 95% CI = [0.9989, 0.9993]), indicating
strong agreement between the methods. Importantly, even with this
alternative calculation, ReHo still exhibits a weaker correlation with
long-range functional connectivity (r(358) = 0.31, p = 3.5201e-08, 95% CI
= [0.20, 0.41]) compared to GWFC (r(358) = 0.67, p = 7.5282e-49, 95% CI
= [0.61, 0.73]), supporting the robustness of our original result.

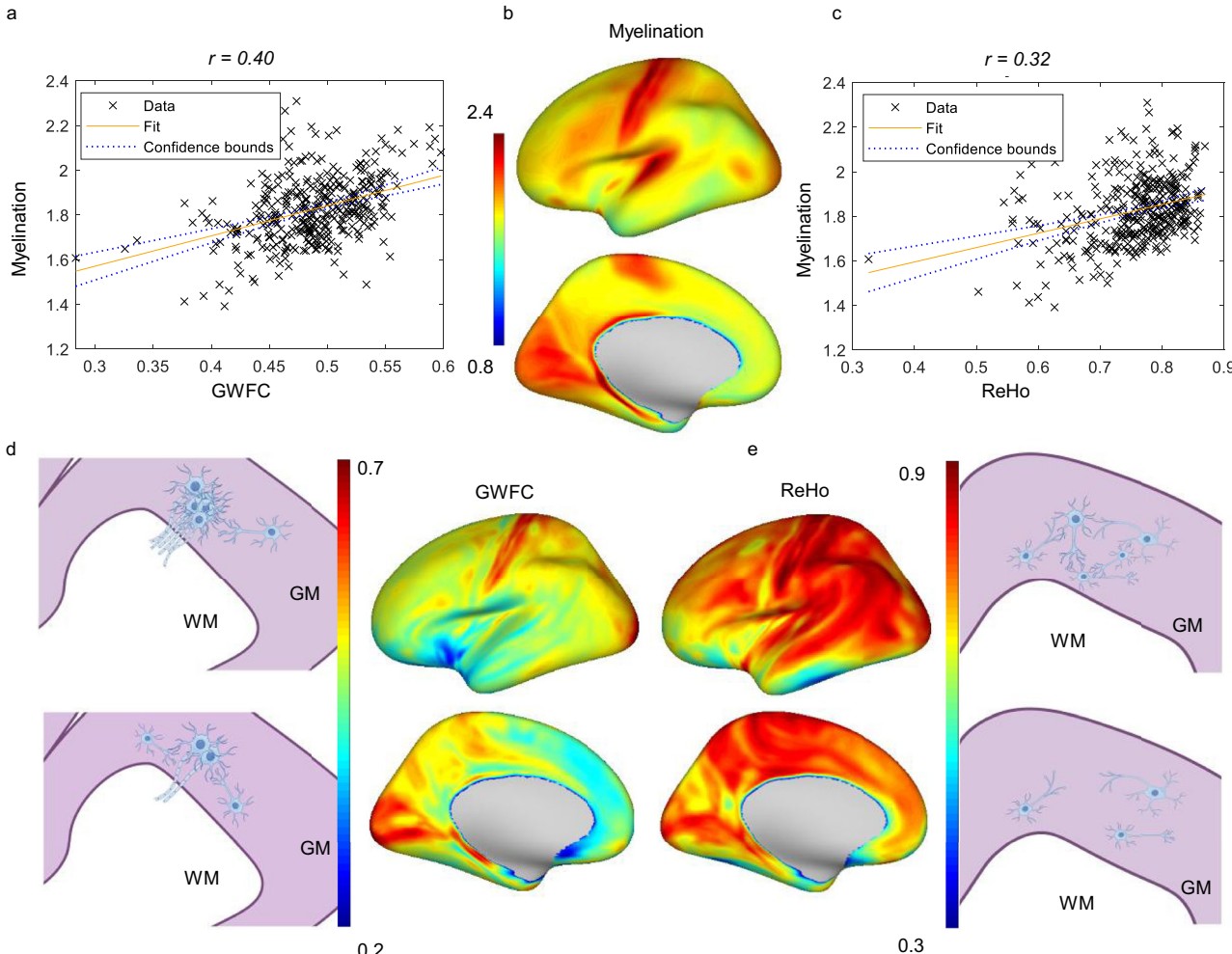

**Fig. 2 | Correlations between myelination, GWFC, and ReHo across the cortex.**
**a** A scatter plot showing the positive correlation (r(358) = 0.40, p = 2.7470e-15, 95% CI = [0.31, 0.48]) between GWFC and myelin content. **b** Spatial distribution of myelin content. **c** A scatter plot showing a weaker positive correlation (r(358) = 0.32, p = 2.9541e-10, 95% CI = [0.23, 0.41]) between ReHo and myelin content. **d** Spatial distribution of GWFC and corresponding anatomical diagrams. **e** Spatial distribution of ReHo and corresponding anatomical diagrams. Each point in the scatterplot represents a brain region, and the fitted line depicts the general relationship between the two measured metrics. Two-sided Pearson correlation was used to assess the association between variables. For the distribution maps, the color represents the metric value (blue = low, red = high). These diagrams in **d**, **e** are intended to conceptually illustrate how different orientations of functional connectivity, perpendicular for GWFC and tangential for ReHo, may reflect varying underlying anatomical and myelin characteristics. The top and bottom schematics, aligned with the color bars, correspond to regions with high and low GWFC or ReHo values, respectively. Some elements in **d**, **e** were created with Figdraw. *GWFC* gray-white matter functional connectivity; ReHo = regional homogeneity.

To further evaluate the spatial nature of long-range connectivity captured by this approach, we generated voxel-wise WM FC maps for two representative GM seed clusters exhibiting the highest GWFC values. As shown in Figure S4, these spatial maps illustrate widespread correlations between GM seeds (in somatosensory and visual cortices) and distributed WM regions across the brain, consistent with known projection pathways such as corticospinal tracts and posterior thalamic radiations.

**Relationship between GWFC and Sensorimotor-association axis**
Figure 4 shows the significant negative correlation (r(358) = −0.41, p = 2.3242e-16, 95% CI = [−0.50, −0.32]) between the spatial maps of GWFC and the sensorimotor-association axis. This axis, originally reported by a previous study[26], represents a principal gradient of cortical organization derived from various techniques, including neuroimaging data, evolutionary and developmental markers, meta-analysis markers, and gene expression profiles. Given the nature of the sensorimotor-association axis, higher GWFC values are associated with lower-order brain functions, such as those involved in sensory and motor processing, whereas lower GWFC values correspond to integrative higher-order functions, such as abstract thinking, memory, and language.

**Relationship between GWFC and neurotransmitter receptors**
Neurotransmitter receptor distribution maps, derived from a recently compiled multimodal atlas of the human brain[30] were used to investigate the neurochemical correlates of GWFC. 35 out of 48 neurotransmitter receptors show significant correlations with GWFC (p < 0.05, Bonferroni correction). Figure 5 shows the three neurotransmitter receptors that exhibit the highest positive correlation with GWFC distribution, serotonin 1 F, noradrenergic beta1, and histamine 2, as well as the three receptors with the highest negative correlation, serotonin 3a, 1a, and 7. These receptors are associated with a diverse range of physiological and cognitive functions. Specifically, serotonin 1 F is involved in pain and migraine modulation[31,32], noradrenergic beta1 plays a role in cardiac function regulation[33], and histamine 2 is linked to gastric acid secretion[34]. In contrast, serotonin 3 A[35,36] is related to nausea and anxiety regulation, serotonin 1 A is associated with

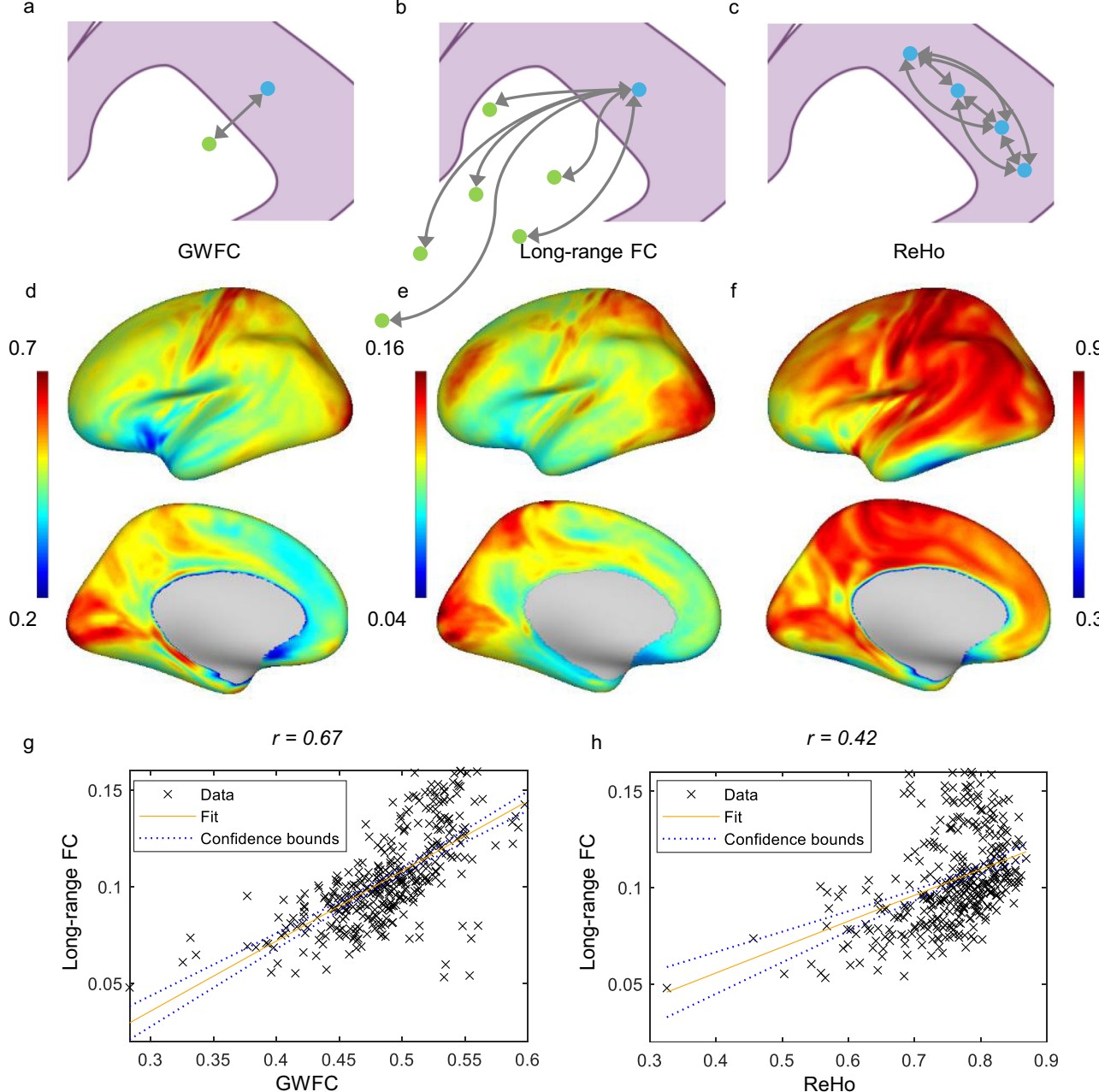

**Fig. 3 | Correlations of GWFC and ReHo with long-range FC. a–c** Illustration of GWFC, long-range functional connectivity, and ReHo. The blue points are on the mid-thickness cortical surface. The green points are in white matter. Some elements in this figure were created with Figdraw. **d–f** Spatial distributions of GWFC, long-range FC, and ReHo. The color represents the metric value (blue = low, red = high). **g** The scatter plot demonstrates a significant positive correlation (r(358) = 0.67, p = 7.5282e-49, 95% CI = [0.61, 0.73]) between GWFC and long-range FC. **h** Relationship between ReHo and long-range FC, showing a positive but weaker correlation (r(358) = 0.42, p = 3.7826e-17, 95% CI = [0.34, 0.51]). Each point in the scatterplot represents a brain region, and the fitted line depicts the general relationship between the two measured metrics. Two-sided Pearson correlation was used to assess the association between variables. GWFC gray-white matter functional connectivity, ReHo regional homogeneity; long-range FC long-range functional connectivity.

mood and anxiety regulation[37], and serotonin 7 contributes to circadian rhythms and mood regulation[38].

## Distribution of GWBPR

As shown in Fig. 6, the GWBPR shows an inverse relationship with GWFC and is negatively correlated with the myelin map (r(358) = −0.32, p = 2.2825e-09, 95% CI = [−0.42, −0.22]) and positively correlated with the sensorimotor-association axis (r(358) = 0.41, p = 5.9262e-15, 95% CI = [0.32, 0.50]). This indicates that regions with higher power coupling across the boundary are more likely to be associated with higher-order

cognitive functions and engage less myelinated white matter tracts. GWBPR also shows significant negative correlations with GWFC (r(358) = −0.68, p = 1.2889e-45, 95% CI = [−0.73, −0.61]) and long-range FC (r(358) = −0.73, p = 4.3120e-56, CI = [−0.78, −0.68]), as shown in Fig. 7. To further examine the functional relevance of GWBPR, we explored its relationship with neurotransmitter receptor distribution. Specifically, we analyzed the correlations between GWBPR and the six receptors that showed the strongest positive and negative associations with GWFC (as identified in Fig. 5). As shown in Figure S5, GWBPR exhibits an inverse correlation pattern with these receptors compared to GWFC.

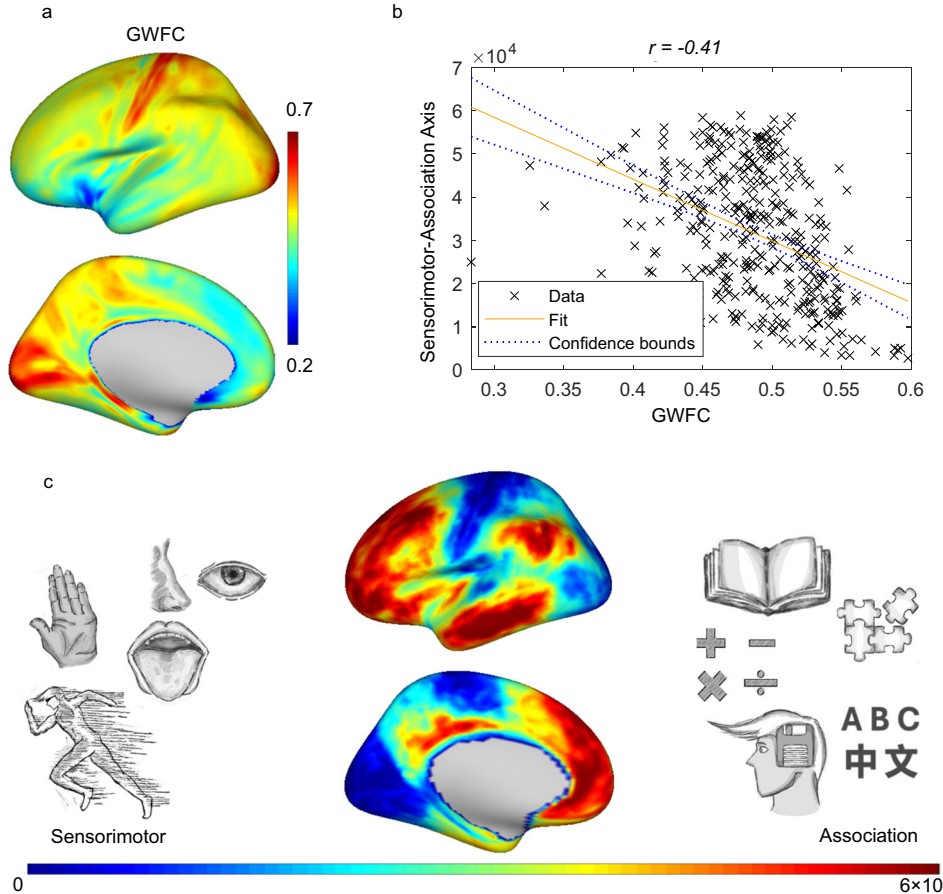

**Fig. 4 | Relationship between GWFC and the sensorimotor-association axis.**
**a** The spatial distribution of GWFC across the cortical surface. The color represents the metric value (blue = low, red = high). **b** A scatter plot illustrates the relationship between GWFC and the sensorimotor-association axis, with a negative correlation coefficient (r(358) = −0.41, p = 2.3242e-16, 95% CI = [−0.50, −0.32]). Two-sided Pearson correlation was used to assess the association between variables. Each point represents a cortical region, and the negative trend indicates that higher GWFC values correspond to sensorimotor functions (lower on the axis), while lower GWFC values are associated with regions that have higher-order association functions. (c) A map of the sensorimotor-association axis is shown, with a gradient spanning from sensory and motor functions (blue) to higher-order association regions (red). GWFC gray-white matter functional connectivity.

## Development of GWBPR over age

The average distribution of GWBPR, calculated based on the HCP-D dataset, is displayed in Fig. 8a. Visual inspection reveals that the late childhood and adolescent group shows widespread lower GWBPR compared to young adults. Statistical comparisons between the two groups were not performed due to differences in scanners, imaging protocols, and preprocessing pipelines. However, within the HCP-D group, a region-by-region analysis of the change in GWBPR with age was conducted. As shown in Fig. 8b, significant correlations ($p < 0.05$, Bonferroni correction) were identified in 107 out of 180 regions defined by the MMP1.0 atlas, all showing positive correlations between age and GWBPR. Figure 8c-j highlights the top 8 regions with the highest correlations, including the ventral visual complex, middle insular area, TG dorsal area, parahippocampal 1 area, anterior agranular insular complex, posterior OFC complex, 47 s area, and TE2 posterior area. These regions are primarily distributed around the temporal, insular, and orbitofrontal cortices and are involved in various higher-order functions, such as visual processing, memory encoding, emotional regulation, social cognition, and decision-making.

## Discussion

We have introduced two functional metrics, GWFC and GWBPR, which measure the fidelity and power coupling of BOLD signal variations across the GM-WM boundary. Our results reveal that GWFC is more strongly associated with myelination, long-range FC, and lower-order functions, while GWBPRs demonstrate a strong association with higher-order cognitive regions.

Our results indicate that GWFC, which represents functional connectivity perpendicular to the GM-WM boundary, correlates more strongly with myelin content than ReHo, which represents connectivity tangential to the boundary. This difference may reflect the distinct underlying biological connections captured by these metrics. GWFC may be more influenced by axonal pathways that cross the GM-WM boundary, which are often more heavily myelinated to facilitate efficient signal transmission. Supporting this, a diffusion MRI study[39] demonstrated that the principal diffusion orientation in the cortex aligns perpendicular to the GM-WM boundary, consistent with the organization of myelinated axons. By contrast, ReHo captures more local synaptic interactions[40]. Therefore, its association with myelin content is naturally weaker.

Based on this evidence, we propose that the distribution of GWFC is strongly influenced by the myelin profiles at different boundary locations. This effect could arise from the level of myelination itself or the specific organization of myelinated axons. First, higher levels of myelination, particularly in projection fibers, could enhance the fidelity and speed of signal transmission, leading to greater GWFC in relevant boundary locations. Second, the organization of these myelinated axons also plays a crucial role. The same diffusion MRI study[39] revealed that within superficial white matter regions containing

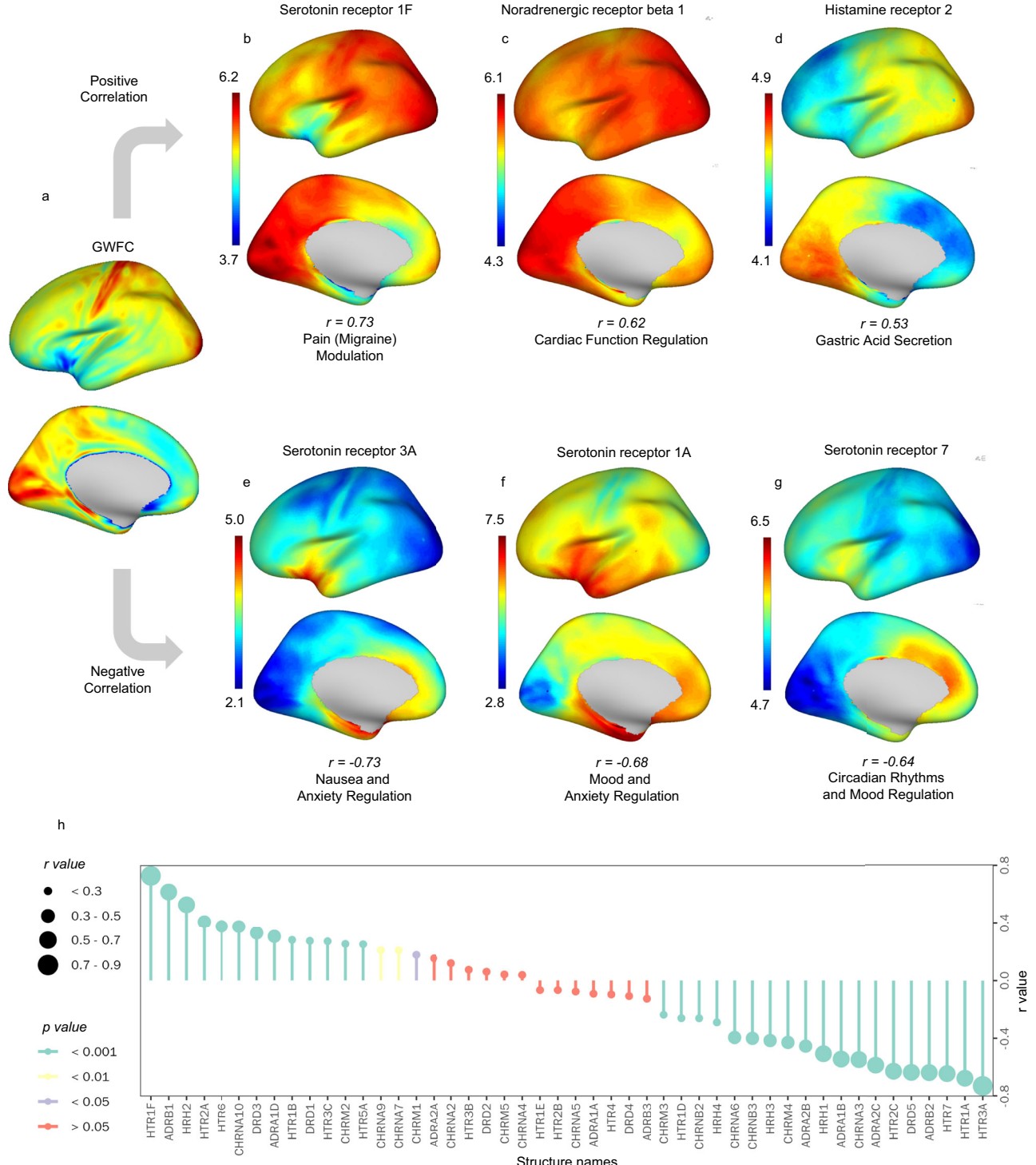

**Fig. 5 | Correlations between GWFC and neurotransmitter receptor distributions. a** The GWFC distribution is shown across the cortex. **b**–**d** The spatial maps display the top three neurotransmitter receptors positively correlated with GWFC. The functional labels under each receptor represent well-established associations from prior literature. **e**–**g** The spatial maps depict the top three neurotransmitter receptors negatively correlated with GWFC. The functional labels under each

receptor represent well-established associations from prior literature. **h** *r* and *p* values of all correlations. Two-sided Pearson correlation was used to assess the association between variables. The p-values have been adjusted for multiple comparisons using the Bonferroni correction. The color in **a**–**g** represents the metric value (blue = low, red = high). GWFC gray-white matter functional connectivity.

---

U-fibers and association fibers, the orientation of fibers tends to be more complex and variable. In these regions intersecting signals[41] potentially mix or cancel each other out, reducing their correlation with cortical activity. In contrast, projection fibers are characterized by more consistent and coherent orientations, often aligning

perpendicular to the GM-WM boundary, supporting more synchronized signal transmission across the boundary.

Although ReHo is less strongly correlated with myelin than GWFC, the correlation remains significant. This could be due to the fact that myelination facilitates efficient signal transmission not only in

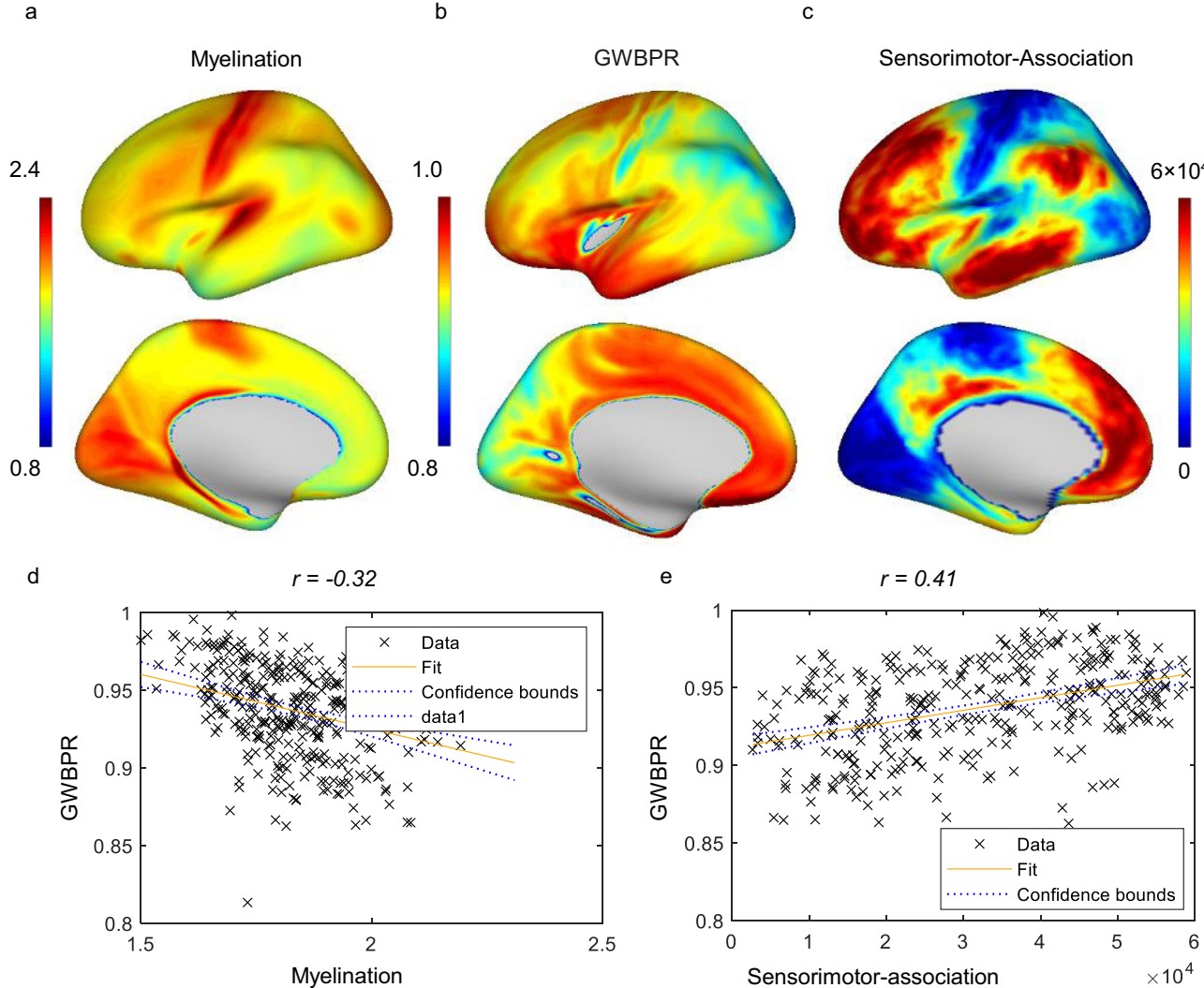

**Fig. 6 | Relationship between GWBPR, myelination, and sensorimotor-association axis. a–c** The cortical distributions of myelination, GWBPR, and the sensorimotor-association axis are shown across the brain. The color represents the metric value (blue = low, red = high). **d** A negative correlation (r(358) = −0.32, *p* = 2.2825e-09, 95% CI = [−0.42, −0.22]) is observed between GWBPR and myelin content. **e** A positive correlation (r(358) = 0.41, *p* = 5.9262e-15, 95% CI = [0.32, 0.50])

is found between the GWBPR and the sensorimotor-association axis. Each point in the scatterplot represents a brain region, and the fitted line depicts the general relationship between the two measured metrics. Two-sided Pearson correlation was used to assess the association between variables. GWBPR gray-white BOLD power ratio.

long-range projection fibers but also in local circuits within the cortex. Even though the connections captured by ReHo are primarily intra-cortical and short-range, the presence of even low levels of myelination may contribute to local signal precision, which in turn may influence fMRI synchrony measures such as ReHo.

While both GWFC and ReHo show correlations with long-range functional connectivity (FC), the stronger correlation observed with GWFC suggests that it captures additional information relevant to the integration of signals across the GM-WM boundary and to distant regions. Notably, both long-range FC and GWFC exhibit particularly high values in sensorimotor regions compared to other areas of the brain. This finding can be explained by two factors. First, sensorimotor regions are connected by densely packed and heavily myelinated tracts, as demonstrated by diffusion MRI studies[42,43], which report consistently high fractional anisotropy (FA) over the entire tracts. These high FA values indicate well-organized fiber structures with minimal crossing or interference from other pathways, leading to signal transmission that is less contaminated by surrounding neural noise. Consequently, the increased signal-to-noise ratio (SNR)

contributes to the higher GWFC and long-range FC observed in these areas. Second, the higher degree of myelination in these tracts enhances the strength and accuracy of functional connectivity, facilitating more synchronized communication even between distant brain regions. Therefore, the combination of well-organized fiber orientation and high myelination allows sensorimotor tracts to maintain stronger correlations of GWFC with long-range FC in these regions.

The significant correspondences between GWFC and certain neurotransmitter receptor distributions, particularly those associated with lower-order functions, may offer clues to the potential neuro-chemical correlates of GWFC. The pattern of agreement suggests that regions with high GWFC may be involved in unimodal signal trans-mission supported by neurotransmitter systems crucial for basic physiological functions. This aligns with the idea that lower-order functions require rapid and accurate information flow, often necessi-tated by heavily myelinated fibers, thereby enhancing the synchroni-zation and functional connectivity in these regions.

In contrast, receptors that show negative correlations with GWFC are associated with higher-order functions and pathologies. Higher-

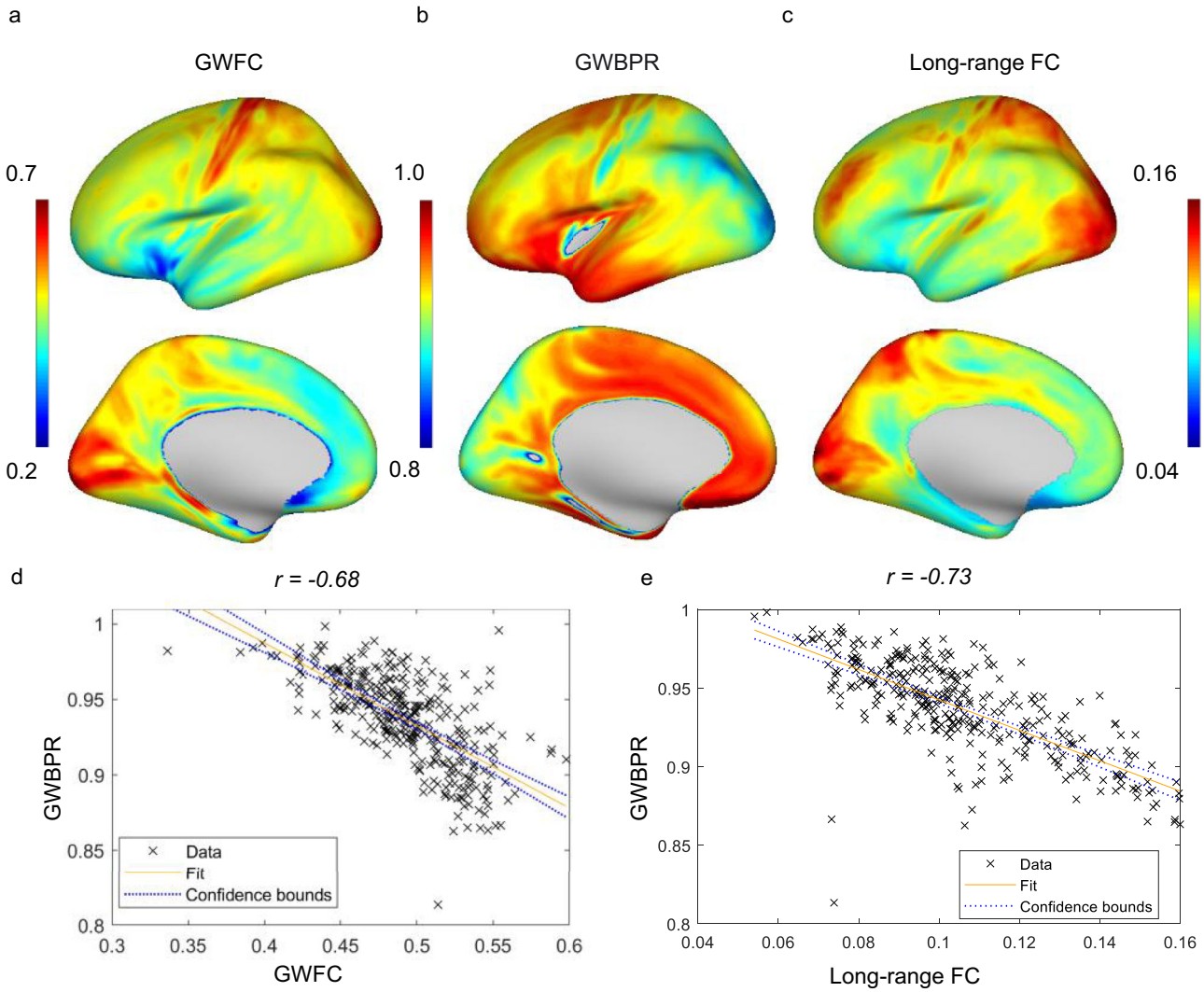

**Fig. 7 | Relationship between GWBPR, GWFC, and long-range FC. a–c** The cortical distributions of GWFC, GWBPR, and the long-range FC are shown across the brain. The color represents the metric value (blue = low, red = high). **d** A negative correlation (r(358) = −0.68, p = 1.2889e-45, 95% CI = [−0.73, −0.61]) is observed between GWBPR and GWFC. **e** A negative correlation (r(358) = −0.73, p = 4.3120e-56, CI = [−0.78, −0.68]) is found between the GWBPR and long-range FC. Each point in the scatterplot represents a brain region, and the fitted line depicts the general relationship between the two measured metrics. Two-sided Pearson correlation was used to assess the association between variables. GWFC gray-white matter functional connectivity; GWBPR gray-white BOLD power ratio; long-range FC long-range functional connectivity.

order regions that process complex cognitive and emotional functions often require flexible, integrative signal transmission[26] rather than rapid and precise transfer[44]. The slower signal transmission and more diverse, less uniform pathways might be beneficial for fine-tuning and complex information processing.

GWBPR is not an absolute measure of BOLD signal amplitude in WM alone but a relative measure between a WM point and its paired GM point across the GM-WM boundary. It reflects the gradient or contrast in signal power, an approximate indicator of local energy demand or metabolic activity[24], between the two tissue types at each location. Our findings suggest that regions with lower GWBPR values tend to exhibit higher levels of myelination. This aligns with the understanding that heavily myelinated fibers associated with rapid and accurate signal transmission[45], as observed in sensorimotor regions, in general produce weaker BOLD effects. The dense and highly organized myelin sheaths that wrap these axons not only reduce signal loss but may also lower the energy demands[46] for signal transmission through highly insulated channels[47]. Consequently, this efficient transmission process in the nearby WM, along with the relatively simple and

unimodal nature of the transmitted signals, leads to the sharper contrast of signal power (reduced GWBPR) observed in these regions. In contrast, regions involved in higher-order processing are often associated with higher GWBPR, reflecting a different type of signal propagation that may not prioritize rapidity but a more balanced energetic profile across the boundary in support of diverse, multimodal processing[48], and may indirectly relate to a gradient in vascularization or neurovascular coupling between the two tissues. While it is well established that energy demand is generally supported by vascularization, we do not have direct evidence regarding regional differences in vascular density across the boundary.

The observed age-related increase in the GWBPR, particularly in regions associated with higher-order functions, suggests that as these regions mature, they may prioritize increased BOLD power to support the greater metabolic demands of complex processing. This observation aligns with the developmental trajectory of higher-order cognitive functions. Unlike sensorimotor functions, which generally reach full maturation between ages 9 and 12[27], higher-order functions continue to mature through adolescence and into early adulthood[49–52].

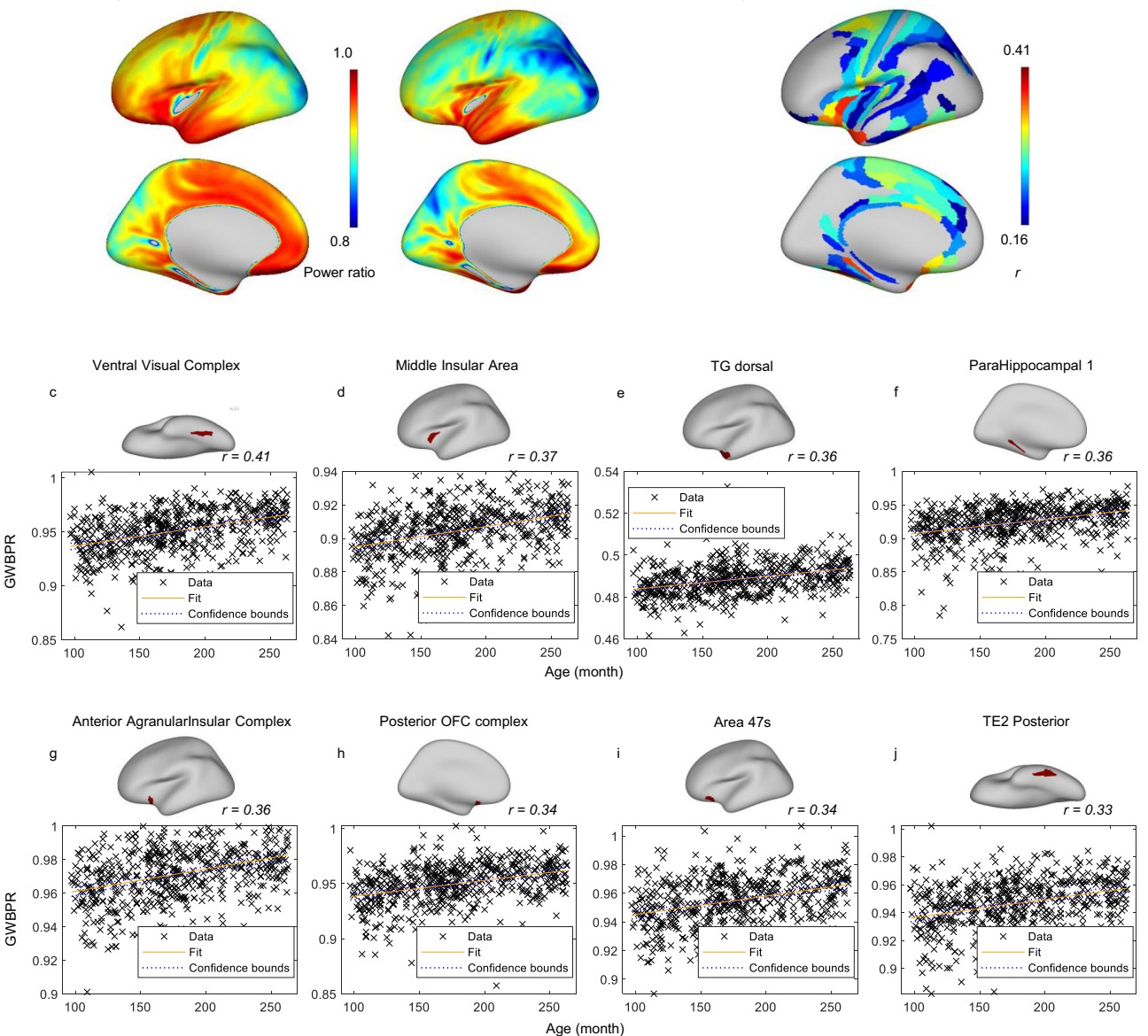

**Fig. 8 | Age-related variations in GWBPR across the cortex. a** Comparison of GWBPR Between Age Groups (HCP-Y vs. HCP-D). The color represents the metric value (blue = low, red = high). **b** Significant correlation between GWBPR and age in the HCP-D cohort: a spatial map (ROI-wise) of the correlation between GWBPR and age is presented, with red colors of each ROI indicating stronger positive correlations. Two-sided Pearson correlation was used to assess the association between variables. Regions showing significant correlations ($p < 0.05$, Bonferroni correction) are displayed. **c–j** Scatter plots showing the relationship between GWBPR and age for the top eight regions that show the highest correlation with age. Each point in the scatterplot represents a brain region, and the fitted line depicts the general relationship between the two measured metrics. The statistics for the eight

comparisons are: **c** r(178) = 0.41, p = 2.6478e-22, 95% CI = [0.33937, 0.47604]. **d** r(178) = 0.37, p = 1.5498e-17, 95% CI = [0.29524, 0.43721]. **e** r(178) = 0.36, p = 4.8513e-17, 95% CI = [0.29028, 0.43281]. **f** r(178) = 0.36, p = 7.1917e-17, 95% CI = [0.28856, 0.43128]. **g** r(178) = 0.36, p = 2.1183e-16, 95% CI = [0.28377, 0.42702]. **h** r(178) = 0.34, p = 4.2285e-15, 95% CI = [0.27009, 0.41484]. **i** r(178) = 0.34, p = 1.2476e-14, 95% CI = [0.265, 0.41028]. **j** r(178) = 0.33, p = 4.1685e-14, 95% CI = [0.25921, 0.40511]. The p values have been adjusted for multiple comparisons using the Bonferroni correction. GWBPR gray-white BOLD power ratio, HCP-Y human connectome project young adult database, HCP-D human connectome project development database, TG Temporal Gyrus, OFC orbitofrontal cortex.

Consequently, the maturation of higher-order regions is associated with the greater BOLD demands observed, potentially supporting their increasing functional and structural complexity during those ages. It is worth noting that although low-order functions, better captured by GWFC, are largely mature by late childhood, we still observed significant age-related increases in GWFC within certain regions, as shown in Figure S6. However, the spatial extent of these changes is relatively limited, which is consistent with our expectations.

Our findings also suggest promising avenues for future research using GWFC and GWBPR as markers of structure-function coupling at

the gray-white mfatter boundary. Given the unique microstructural properties of the boundary, where diverse axonal projections converge and myelination gradients shift, these functional measures may be sensitive to early changes in development, learning, and neuropathology. For example, longitudinal studies could explore whether GWFC is altered in the early stages of demyelinating diseases (e.g., multiple sclerosis) or neurodegenerative conditions (e.g., Alzheimer's disease), potentially serving as an early functional indicator before widespread white matter disruption. Similarly, boundary metrics may provide insight into experience-dependent plasticity during learning

or cognitive training, particularly in association cortices where myelination is prolonged into adulthood. Future work integrating these boundary measures with structural imaging across time could help clarify whether functional changes at the GM-WM interface precede, accompany, or follow underlying anatomical changes.

A key limitation of this study is the interpolation used to estimate BOLD signals in WM voxels that are 1 mm away from the GM-WM boundary, given that the fMRI resolution is 2 mm. This interpolation could inadvertently include signals from nearby subcortical structures, potentially contaminating WM signal measurements. Although we applied a mask to exclude subcortical structures by setting their values to zero, this may have introduced extremely low GWBPR values near zero, which may not accurately reflect true power alteration. The exclusion of subcortical structures also prevented us from examining the functional role of subcortical structures, such as the thalamus and basal ganglia, which are known to act as key relays in brain communication. This issue is a limitation of current imaging techniques, as all surface-based fMRI studies face similar challenges. Nonetheless, future work should aim to utilize higher-resolution data and incorporate subcortical structures into the analysis to provide a more complete picture of large-scale signal integration.

A related concern is that, due to this 1 mm distance, partial volume effects may allow GM signals to contaminate WM measurements. To investigate this, we conducted a control analysis using a more conservative sampling distance of 4 mm beneath the boundary, equivalent to two fMRI voxels, where contamination from the GM signal is expected to be minimal. As shown in Figure S7 and Figure S8, the spatial distributions of both GWFC and GWBPR remained consistent with our original findings, and their correlations with the myelin content were preserved. These results support the robustness of our conclusions and suggest that they are not driven by partial volume contamination.

Another limitation of our analysis is that the neurotransmitter receptor maps were derived from independent datasets and therefore are not directly linked to the fMRI data used to compute GWFC. As a result, the observed correlations may not fully capture the true biological coupling between receptor distribution and functional dynamics.

In conclusion, our findings provide a comprehensive view of how myelination, functional connectivity, and neurotransmitter distribution shape the functional architecture of the GM-WM boundary of the brain. GWFC and GWBPR offer complementary insights into the fidelity and magnitude variation of signal propagation across the GM-WM boundary, with implications for understanding both healthy brain function and potential disruptions in neurodevelopmental and neurodegenerative disorders.

## Methods

### Ethics Statement
The data involved in this research are publicly available and have been previously approved for use by the Washington University Institutional Review Board. All participants provided written informed consent to participate in this study. The authors did not collect any new data involving human participants.

### Dataset
We selected 687 entries from the Human Connectome Project Young Adult (HCP-Y)[29] repository (comprising 335 males and 352 females, all between the ages of 22 and 35 years) and 571 subjects from HCP development (HCP-D)[28] repository (259 males and 312 females, between ages of 8 and 21 years), adhering to criteria that included the completeness of a 3 T scan and associated physiological data, along with adequate data quality. We have included only the relevant MRI modalities that were pertinent to our study from these databases. The sex of participants was recorded based on self-report in the HCP dataset. No sex- or gender-based analyses were conducted in this study.

For HCP-Y, the imaging protocols are detailed elsewhere[53] and were performed with 3 T Siemens Skyra scanners. Resting-state scans were acquired using multiband gradient-echo EPI sequences. Each session consisted of two runs of scans with opposing phase encoding directions and lasted 14 min and 33 sec, with parameters TR = 720 ms, TE = 33.1 ms, and an isotropic voxel resolution of 2 mm, totaling 1200 volumes. Concurrent recordings of physiological responses, such as respiration and heartbeat, were captured during fMRI scans. Additionally, T1-weighted images were obtained using a single-echo MPRAGE sequence with a TR of 2400 ms, TE of 2.14 ms, and voxel dimensions of 0.7 mm isotropic. T2-weighted images were obtained using a 3D T2-SPACE sequence with a TR of 3200 ms, TE of 565 ms, and voxel dimensions of 0.7 mm isotropic.

For HCP-D, the imaging protocols are more thoroughly described elsewhere[54]. Briefly, scans were executed on Siemens 3 T Prisma scanners with 32-channel head coils. The resting-state fMRI protocol involved four runs with opposing phase encoding directions, each 6 min and 41 sec, with TR = 800 ms, TE = 37 ms, voxel dimension = 2 mm isotropic, and a total of 488 volumes for each run, while physiological parameters were also documented. T1-weighted images were obtained using a multi-echo MPRAGE sequence, with a TR of 2500 ms, TEs of 1.8/3.6/5.4/7.2 ms, and voxel dimensions of 0.8 mm.

### Preprocessing
The datasets acquired through ICA-FIX, which have regressed out white matter signals, were not used in this study[55]. Our approach was to employ 'uncleaned' images that underwent only the Minimal Preprocessing Pipelines[56]. Briefly, T1-weighted images were nonlinearly coregistered to MNI space using FNIRT[57], with subsequent processing via the Freesurfer suite, resulting in mesh surface profiles for different cortical layers, volumetric and surface parcellations, as well as voxel (volumetric) and vertex (surface) -wise BOLD time series[58]. This study utilizes two mesh surfaces generated by Freesurfer: the WM surface, marking the boundary between GM and WM; and the mid-thickness cortical surface, located midway between the pial and white matter surfaces. These surface profiles record the MNI coordinates of each vertex and the connectivity between vertices, which are organized into triangular faces. Meanwhile, the T2-weighted images were aligned with the native T1-weighted images using 6 degrees of freedom (DOF) and subsequently registered to MNI space along with the T1-weighted images. The fMRI processing encompassed the removal of head movement artifacts, correction of distortions from susceptibility effects using FSL based on the two runs of data with opposite-phase encoding directions, and then nonlinear registration to MNI space. Further processing steps included regressing out confounding variables including 12 head movement parameters and physiological fluctuations, modeled by the RETROICOR technique[59]. This preceded the application of linear trend corrections and temporal filtering using a band-pass filter covering the frequency range of 0.01–0.1 Hz.

### GWFC, GWBPR, and ReHo calculation
In surface analysis, the cortical surfaces of different layers are spatially aligned, meaning that each vertex on one surface corresponds to a vertex on the other surface. For example, a vertex on the pial surface has a corresponding vertex on the white matter surface, and those two represent the same location in the cortex, but on different layers, allowing for consistent comparisons and measurements across surfaces. The line connecting two corresponding vertices is approximately aligned with the normal direction of both surfaces.

In our study, two functional contrasts, including GWFC and GWBPR, were calculated for each vertex $v_i$ on the boundary surface, based on the BOLD signals from a GM point and a WM point on either side of the boundary. As shown in Fig. 1c, the GM point is defined by the

vertex on the mid-thickness surface, which spatially corresponds to $v_i$. The WM point lies along the normal line extending from $v_i$ to WM, positioned 1 mm away from the boundary surface. Note that the fMRI voxel size in this study is 2 mm isotropic, and thereby trilinear interpolation was applied to estimate the BOLD signal at sub-voxel precision at those points and vertices. It is important to note that, as some subcortical structures are physically close to certain boundary areas, such as the putamen, the interpolation process could introduce GM signals into the measured signal of WM points. To eliminate this, we set the fMRI intensity of all subcortical structures to zero before calculating GWFC and the GWBPR. The GWFC is defined as Pearson's correlation between each pair of GM and the WM points that are aligned perpendicular to the surface.

$$GWFC(v_i) = corr(BOLD_G, BOLD_W)$$

The GWBPR is defined as the ratio of fALFF at the WM point to the fALFF at the GM point, where fALFF is itself calculated as the ratio of the power within the low-frequency range (0.01–0.1 Hz) to the total power of all frequencies detectable in the signal.

$$GWBPR_{(v_i)} = \frac{fALFF_W}{fALFF_G}$$

ReHo for each vertex is calculated based on the functional connectivity among a group of vertices on the mid-thickness surface, including the vertex itself and its directly connected neighboring vertices, as shown in Fig. 1d. Such connectivity was quantified by Kendall's coefficient of concordance (KCC)[60].

$$ReHo(v_i) = KCC\left(\left\{BOLD_{Neighbors\ of\ v_i}\right\}\right)$$

In summary, while GWFC and the GWBPR represent contrasts in connectivity along the perpendicular direction to the surface, ReHo captures connectivity along the tangential direction to the surface.

## Myelin map
We quantified myelin content using the T1-weighted (T1w) to T2-weighted (T2w) ratio method, leveraging data obtained from the HCP young adults. This approach is grounded in the principle that the ratio of T1w to T2w signal intensities in brain images correlates with myelin content, as demonstrated by Glasser and Van Essen[61] and further elaborated by Ganzetti et al.[62]. The surface myelin map for each subject was obtained from the HCP database, where the T1w/T2w ratio was calculated for each vertex, producing a surface map that illustrates myelin distribution and density across different cortical regions. It is important to note that myelin content was not measured at a single vertex; instead, for each vertex, a ribbon was reconstructed, covering the entire space between the pial and white matter surfaces. The T1w/T2w values were sampled and averaged from multiple points within this ribbon, meaning the measured myelin content represents the average across all cortical layers.

## Long-range FC
One of our hypotheses is that connections perpendicular to the GM-WM boundary are likely to be involved in long-distance information exchange through WM tracts. To test this, a long-range functional connectivity (FC) measure was generated for each vertex on the boundary. This was achieved by first calculating the Pearson correlation coefficients between the BOLD signals of each GM vertex on the mid-thickness cortical surface and all WM voxels (30,932 in total) across the brain, and then averaging these 30,932 coefficients. The WM voxels were selected using a group WM mask, created by averaging the Freesurfer-derived WM parcellations from all participants and applying a threshold of 0.95.

## Sensorimotor-association axis map
The sensorimotor-association axis map was replicated from a previous study[26]. It describes a continuum from primary and unimodal sensory and motor cortices (regions that process basic sensory inputs and control movements) to multimodal areas (which integrate information from multiple sensory modalities), and ultimately to trans-modal association cortices (regions involved in higher-order cognitive functions, such as abstract thinking, decision-making, and social cognition). Briefly, the values that parameterize this map represent an integration of multiple neurobiological properties measured from various techniques, including neuroimaging data, evolutionary and developmental markers, meta-analysis markers, and gene expression profiles. It was reconstructed by first ranking each vertex on a property map according to its relevance to cortical function, with lower-order properties at one end and higher-order tasks at the other. For example, the gene expression map was ranked based on the gradient from vertices expressing genes related to sensory functions to those associated with higher-order cognitive functions. All different rank-order maps (10 in total) were then averaged to reconstruct the sensorimotor-association axis map used in this study.

## Neurotransmitter receptors
The receptor maps used in this study were replicated from a previous work[30,63]. Briefly, these maps were generated using Gaussian process regression to predict mRNA expression (Allen Human Brain atlas http://www.brain-map.org) levels of neurotransmitter receptors across the cortical surface. This method allowed for comprehensive receptor distribution maps, even in regions where direct receptor binding measurements were not available. To ensure the accuracy of these maps, the predicted mRNA expression levels were validated against positron emission tomography (PET) binding potential data collected from 30 healthy individuals. The strong correlation between the predicted receptor distributions and PET data provided external validation of the receptor maps. In total, 48 receptor maps were produced, spanning multiple neurotransmitter systems, including serotonin, dopamine, noradrenaline, and acetylcholine receptors. These receptor maps were then used in the present study for multimodal comparisons to examine the neurochemical basis of the distribution of boundary functional contrast.

## Similarities between spatial maps
A group-averaged surface map was reconstructed for each of the measurements mentioned above. The similarities between these spatial maps were assessed using Pearson's correlation. Specifically, by using the multi-modal parcellation (MMP) 1.0 atlas[64], each map was segmented into 360 regions (180 per hemisphere), thereby converting the map into a vector of length 360, with each element corresponding to the mean measurement within a region. Pearson's correlation was calculated to evaluate the spatial similarity between these maps across the 360 regions.

## Reporting summary
Further information on research design is available in the Nature Portfolio Reporting Summary linked to this article.

# Data availability
The MRI data used in this study are available in the HCP database https://www.humanconnectome.org/ The source data used for generating the Figures in this study are provided in the Supplementary Information/Source Data file. Source data are provided with this paper.

# Code availability
The code used for generating the two metrics in this study is available on GitHub at: https://github.com/geyerou/Gray-White-Matter-Boundary. Detailed instructions for reproducing the key results are

provided in the repository's README file. Other software and toolboxes that are required for running the code include: DPABI: http://rfmri.org/DPABI GIFTI: https://www.nitrc.org/projects/gifti/ HCP workbench: https://www.humanconnectome.org/software/connectome-workbench.

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

## Acknowledgements

This work was supported by the National Institutes of Health (NIH) grants R01 NS113832 (JG), R01 NS129855 (ZD), R21 AG083915 (YG), K01 EB032898 (KS), and T32 EB001628 (SC). Data of young adults were provided by the Human Connectome Project, WU-Minn Consortium (Principal Investigators: David Van Essen and Kamil Ugurbil; 1U54MH091657) funded by the 16 NIH Institutes and Centers that support the NIH Blueprint for Neuroscience Research; and by the McDonnell Center for Systems Neuroscience at Washington University. HCP-Development data reported in this publication were supported by the National Institute of Mental Health of the National Institutes of Health under Award Number U01MH109589 and by funds provided by the McDonnell Center for Systems Neuroscience at Washington University in St. Louis. The HCP-Development 2.0 Release data used in this report came from DOI: 10.15154/1520708. We thank Zhaochen Li for his assistance with the illustrations in this paper.

## Author contributions

M.L.: writing–original draft, visualization, validation, software, methodology, investigation, conceptualization. L.X., S.C., Y.Q., F.G., Y.G. and Z.Z.: investigation. K.G.S.: Software, Investigation, Data curation. A.W.A.: Supervision. Z.D.: supervision, methodology, funding acquisition. J.C.G: writing–review & editing, Supervision, Investigation, Funding acquisition.

## Competing interests

The authors declare no competing interests.
