## [Transparent Peer Review file · Nature Communications]

Functional Contrast Across the Gray-White Matter Boundary

Corresponding Author: Dr Muwei Li

Version 0:

Reviewer comments:

Reviewer #1

(Remarks to the Author)

The manuscript describes a new technique to study functional connectivity across the gray-white matter boundary, including novel results about the relationships among gray-white matter boundary connectivity, long range connectivity, myelination measured via T2:T1, the sensorimotor-association functional axis, and neurotransmitter receptor distribution. Results are also reported regarding age differences during development.

The work is novel and of significance to the field.

Major points:

1. For the developmental comparison, why was only GWBPR reported? How did GWFC look?
2. Causal language, e.g., on page 4 "High values of GWFC indicate the BOLD signal in WM closely follows the time course of neighboring GM." Another example on page 5: "Supporting our hypothesis that GWFC is more involved in the transmission of signals to distant regions away from GM"
 - Could it be that the signal of origin is from GM at a distance away (ie via longer range connections), and it is the neighbouring GM that is closely following the WM signal? I also interpret this phrasing as implying that GM signal is more important than WM signal. This may be fine but if it is not the authors' intention, they may wish to rephrase.
3. Use/interpretation of ReHo results
 - In the comparison between ReHo and GWFC, two variables are changing: The analysis method and the voxels to which the method is being applied. As a result, the interpretation that GWFC provides "unique insights" may be too strong. It could be that the unique insights are related to the use of Pearson's rather than KCC. Did the authors consider a more apples-to-apples comparison such that Pearson's correlation is applied neighbouring voxels tangential to the boundary, in order to evaluate whether the GWFC method is providing unique insights?
4. The finding that GWFC is negatively correlated with the sensorimotor-association axis but positively correlated with long range FC seems on its surface to be somewhat contradictory to me. Association areas are involved in very distributed brain networks and have many long range connections. What might be the mechanism for or explanation of this apparent contradiction?

Minor points

1. I think the title is misleading. I was expecting the manuscript to be about differences in functional contrast across the boundary. Why not use the "connectivity" as the method is described throughout the manuscript?
2. "In addition, due to its structural complexity, the boundary is particularly vulnerable to various diseases and injuries, such as abscesses⁵, metastasis⁶, and diffuse axonal injury⁷"
 - What is meant by "structural complexity"? Is the boundary more complex than gray matter itself?
3. "Despite its importance, the functional interaction across the boundary, which potentially measures signal alterations that occur when it traverses from one type of tissue to another, remains poorly understood."
 - Please revise this sentence. "functional interaction" is not a measurement and it is unclear what "it" is referring to.
4. Figure 2 & 3: how were the scale bar ranges selected for the different measures?

5. Page 7: "Correlation analyses confirm that the myelin map is more strongly associated with GWFC ($r = 0.40$, $p = 2.7470e-15$) than by ReHo ($r = 0.32$, $p = 2.9541e-10$) Is this a meaningful/significant difference in correlation coefficients? How was this evaluated?"

6. Why are statistical corrections for multiple comparisons only introduced on page 10? Is a Bonferroni correction appropriate? Why are the other stats from previous pages not corrected for multiple comparisons?

7. Page 12 refers to Figure 8; should be Figure 7

8. Page 20: Should be KCC not KKC

(Remarks on code availability)

Reviewer #2

(Remarks to the Author)

The purpose of the study is to introduce two novel fMRI-derived metrics, gray-white matter functional connectivity (GWFC) and gray-white BOLD power ratio (GWBPR). These novel metrics are meant to quantify functional relationships at the gray-white matter boundary in the brain.

Additionally, these novel metrics are compared to other common metrics, such as myelin levels and GM regional homogeneity (ReHo). Lastly, a cohort of 8-21-year-olds were used to determine the trajectory of GWFC and GWBPR during maturation.

This is an important and unique study that should be published once the major and minor points below are addressed. The functional measures of the WM-GM boundary may be an important region to identify the timeline for disease/injury or for the timeline of learning, but clarity is needed to better convey the importance of these novel metrics and the limitations of them.

Major points:

- These novel functional measures of the WM-GM boundary may be an important region to identify the timeline for disease/injury or for the timeline of learning. It would be especially interesting to longitudinally study these novel metrics along with structural measures either during disease or learning. It would be interesting to see the progression of GWFC with cognitive aging/dementia or demyelinated diseases to see if this boundary is affected before the rest of the myelinated axon. This is one example of interesting future work that can stem from this, however, the future work section and meaning/future implications of the adoption of these measures in the discussion is rather scarce.

- The main text of the manuscript (not including the methods) does not mention the demographics of the data used. There is only one small mention of "(8-21)" in the introduction when mentioning age findings. Please mention that HCP datasets are used, which ones, and that this data comes from 8-21 and 22-35 year-olds. HCP-D is mentioned in the results but without any context of the datasets, the reader does not even know that openly sourced data is used for the current study. Along with this, it should be stated that the myelin maps and the neurotransmitter maps are both from different datasets and are different from the fMRI datasets. This should be explicitly stated and mentioned as a limitation.

- Figure 5 mentions attribute under each of the 6 neurotransmitter receptors that I'm assuming refer to factors the neurotransmitter receptor is associated with [e.g. Serotonin receptor 1F includes Pain (Migraine) modulation]? This is never explained. Also, please state where these associations were pulled from.

Minor points:

- The abstract does not convey the literature gap (specifically white matter fMRI understanding) that is brought up in the introduction.

- Throughout the paper, GWFC is referred to as GWFC and correlation and connectivity. Similarly, GWBPR is sometimes referred to as power ratio. Please be consistent with using the defined abbreviations instead of introducing different phrases for these measures.

- The introduction mentions that these novel functional metrics complement traditional structural assessments of the GM-WM boundary. Are you just referring to the structural contrast of T1w imaging? If there is more that you're referring to, please expand on this.

- As there are many different methods to quantify myelin, in the introduction, it may be important to briefly mention that the current study uses the T1w/T2w ratio method.

- The sensorimotor-association map should be briefly explained in the introduction or results or figure caption to provide a better idea of what the map is telling us. This would help highlight the importance of GWFC association with it and the

discussion points linking high GWFC with more basic sensorimotor functions, without having to dig deep into the methods to understand what the map was.

- The third-last and second-last paragraphs in the introduction discuss the results and the potential implications of them. These paragraphs and discussions belong in the discussion section, not the introduction.

- Figures 2, 3, 4, and 6 should include letters to refer to in the caption for organization. Similar to what you have done in other figures.

- The figure 2 schematic should be better explained. I.e. state that the top and bottom schematics refer to high and low GWFC/ReHo.

- This is a limitation that should be mentioned in the main text and the potential meaning/result/implication of this limitation in voxels and their size should be mentioned: "The WM point lies along the normal line extending from vi to WM, positioned 1 mm away from the boundary surface. Note that the fMRI voxel size in this study is 2 mm isotropic and thereby trilinear interpolation was applied to estimate the BOLD signal at sub-voxel precision at those points and vertex. It is important to note that, as some subcortical structures are physically close to certain boundary areas, such as putamen, the interpolation process could introduce GM signals into the measured signal of WM points. To eliminate this, we set the fMRI intensity of all subcortical structures to zero before calculating GWFC and the power ratio." Could also mention the limitation of the myelin measure being an average of all cortical layers instead of just the value at the vertex.

- The orientation of intersecting WM fibers is a limitation of the long-range correlation of GWFC and its correlation with WM microstructure, correct? Specifically based on your discussion: "The same diffusion MRI study²⁷ revealed that within superficial white matter regions containing U-fibers and association fibers, the orientation of fibers tends to be more complex and variable. In these regions intersecting signals²⁸ potentially mix or cancel each other out, reducing their correlation with cortical activity. In contrast, projection fibers are characterized by more consistent and coherent orientations, often aligning perpendicular to the GM-WM boundary, supporting more synchronized signal transmission across the boundary."

(Remarks on code availability)

The code could include more comments to make it easier to understand input/output data types and sizes.

Reviewer #3

(Remarks to the Author)

(Remarks on code availability)

The README does not include installation or use instructions. There could be more comments in the code to make it easier to understand and to better understand input/output data types and sizes. There are some comments to get a brief overview of what some code sections are doing.

Reviewer #4

(Remarks to the Author)

This manuscript introduces two new measures of BOLD functional connectivity: gray-white matter FC (GWFC) defined as the BOLD correlation between each pair of points perpendicular to the GW-WM boundary along the brain surface, and the gray-white BOLD power ratio (GWBPR) defined as the ratio of BOLD fALFF between the two points (GM vs WM) of each pair. The authors go on to compare the resulting brain surface patterns for GWFC and GWBPR and also for ReHo with surface patterns for T1w/T2w ratio (a proxy for myelination), neurotransmitter density distributions and sensori-motor association axis. They also examine the evolution of GWBPR with brain development during adolescence.

The ideas are interesting, and the manuscript is written clearly. My utmost concern is the partial volume effects between cortex and subcortical WM, that the authors describe as a limitation but take no specific precautions or sanity checks to convince the reader that this PVE does not drive the results of the paper entirely.

Specifically, the point in WM perpendicular to the surface is taken 1 mm from the WM-GM boundary, for an fMRI acquisition with 2-mm resolution. How was this 1-mm distance chosen? It is crucial that the authors test any potential spatial pattern association of their metrics with cortical thickness and gyrus/sulcus curvature, to ensure the associations are not driven by local brain morphology that would introduce more or less BOLD PVE of GM into WM.

Other major revision points that would go in the direction of reinforcing confidence in the message of the manuscript are listed below. Further minor points follow.

- The authors seem to imply that there is a consistent directionality of information transfer from GM to WM, e.g. "how much of the power of the GM BOLD signal is retained as it traverses the boundary". However, this directionality varies: sometimes the

WM connections are efferent into the GM, e.g. in the case of the visual system, the information comes from the optic nerves into the LGN and is then relayed to the visual cortex via the optic radiations. How does this *actual* directionality impact the analysis and the interpretation? The situation for higher order functions is even more complex in terms of information flowing into or out of the GM and from or into the WM.

- It seems to me that long-range connectivity resembles in fact some sort of seed-based analysis, looking at the connections of the seed in the WM. Why not show such connectivity spatial maps for example seeds in the GM? Also, does it make sense to average over the whole WM for each seed, rather than over voxels significantly correlated with the seed in the WM?

- Related to this point, if the assumption is that myelin transmission is fast and metabolically non-demanding, then the GWFC should be preserved across WM voxels belonging to the same fiber to which the initial WM voxel considered close to the boundary belongs. Showing high GWFC along a whole bundle would increase the confidence that this metric is not heavily affected by PVE in proximity to the cortex.

- Too many statements are made using "plausibly", without literature references to back them up. If there is no clear literature reference supporting these statements, please say so explicitly. This impression of speculative statements is reinforced by the fact that the Introduction already develops substantially on all the findings and interpretations of the paper, which better belong in Discussion alone.

- Why would higher order regions require greater BOLD signal power (i.e. higher ratio of fALFF in the WM point vs fALFF in the matching GM point across the border) ? Is the WM there more vascularized? Do they involve more subcortical regions that act as relays, which are often described as a mixture of GM and WM, with both vascularization and myelin?

- "thereby require a high degree of plasticity, which however may be impeded by myelination". Is there a reference for this? Myelination is actually an active component of brain plasticity.

- The relationship between long range FC and GWBPR could also be reported, to complete the analysis of its association with GWFC and ReHo

- Same suggestion for the relationship with neurotransmitter receptor densities. Why were they correlated to GWFC but not other metrics introduced in the paper? E.g. receptors that show negative correlations with GWFC: do they show positive correlation with the BOLD power ratio instead, as of their association with higher order functions?

- P. 13-14: § "Although ReHo [...] the presence of even low levels of myelination still plays a role in improving the precision and timing of local neural signals, contributing to the significant correlation." What range of timings and delays do the authors mean? The precision and timing of local neural signals, with or without myelin, is likely much faster than the BOLD response and the temporal resolution of the acquisition (TR = 720 or 800 ms).

- P.16: "The dense [...] isolated channels." Can the authors provide a reference supporting this?

Minor:

- The reference to the GM-WM boundary and the information "exchanged" between tissue types is a bit surprising to me. Is this description backed by literature? A simpler picture would be neuronal cell bodies in the gray matter projecting axons into the WM to create synapses with distant areas, but why this notion of information exchange at the boundary?

- P.12: "adolescent group shows widespread reductions in the power ratio compared to young adults": better to phrase this as lower power ratio as compared to young adults, since the data is cross-sectional.

- It is a pity that subcortical regions were masked out, if I understood correctly. Indeed, they act as relays which would have been interesting to study in the context of this work.

(Remarks on code availability)

Version 1:

Reviewer comments:

Reviewer #1

(Remarks to the Author)

The authors have thoroughly addressed the concerns raised. I have two minor observations related to the abstract.

Note age range (8-35, ie not older adults)

The abstract defines the sensorimotor-association axis differently (with a focus on receptors) compared to the results which on page 10, line 194 describes the axis as being derived from various techniques.

(Remarks on code availability)

Reviewer #2

(Remarks to the Author)

The authors have made a significant effort to improve their manuscript based on previous reviewer points. They have addressed all my previous concerns. Congratulations on the important and interesting work.

(Remarks on code availability)

All points addressed

Reviewer #3

(Remarks to the Author)

(Remarks on code availability)

Reviewer #4

(Remarks to the Author)

The authors have put substantial effort in the revision, which is appreciated. There are a few important points that would be worth addressing further:

1. Several reviewers suggested more paralleled analyses of GWFC and GWBPR vs other measures, such as long range FC, ReHo, neurotransmitter densities etc. While the authors have provided all of those in the revised version, it still seems a bit arbitrary which ones go in the main text vs supplementary material. Especially as the correlation of long range FC is stronger with GWBPR (supplementary) than with GWFC (main), and the strengths of correlations with neurotransmitter distributions are very similar (yet GWFC is in main, GWBPR in supplementary). This could suggest the authors made the decision based on what fit their narrative, and while this was most likely not their intention or rationale, it would make the paper stronger if they provided more balanced analyses of the two measures. More importantly, I wonder how strongly anti-correlated GWFC and GWBPR are, and therefore whether they really provide complementary information or are perhaps quite redundant (i.e. one is the inverse of the other).

2. Another point that would be important to understand is the interpretation of low GWFC (so low temporal correlation between GM and WM BOLD timecourses at these boundary points) associated rather with high GWBPR (so high fALFF for BOLD timecourse in WM at that point), and vice versa. The high GWBPR suggests high BOLD power in the expected functional frequency ranges, yet the temporal correlation with the GM BOLD is poor. So could the authors also run cross-correlation of BOLD WM vs GM timecourses for a range of temporal delays, to check if there is a delay between the two which would explain the low GWFC but high GWBPR? And would accounting for that temporal delay of the BOLD response in WM boost the GWFC for those regions?

3. This reviewer had understood the definition of the GWBPR correctly the first time around, there was no confusion involved. My question was whether regions involved in higher order functions had higher GWBPR because of higher vascularization in the WM there, but there seems to be no available information on WM vascularization for different bundles and networks. As a side note, given that the ratio is that of $fALFF(WM)/fALFF(GM)$, I wonder if the measure should not better be called WGBPR than GWBPR.

(Remarks on code availability)

Version 2:

Reviewer comments:

Reviewer #4

(Remarks to the Author)

I have no further comments or suggestions to make regarding this manuscript.

(Remarks on code availability)

Reviewer #1 (Remarks to the Author):

The manuscript describes a new technique to study functional connectivity across the gray-white matter boundary, including novel results about the relationships among gray-white matter boundary connectivity, long range connectivity, myelination measured via T2:T1, the sensorimotor-association functional axis, and neurotransmitter receptor distribution. Results are also reported regarding age differences during development.

The work is novel and of significance to the field.

We sincerely thank you for your positive evaluation of our work.

Major points:

1. For the developmental comparison, why was only GWBPR reported? How did GWFC look?

Thank you for this insightful question. We initially focused on age-related changes in GWBPR because, in the HCP-D cohort (ages 8–21), higher-order functions, which are more prominently reflected by GWBPR, are still undergoing significant development. In contrast, lower-order sensorimotor functions, which are better captured by GWFC, are largely mature by late childhood. Therefore, we expected more pronounced developmental effects in GWBPR than in GWFC. To address your question directly, we have now examined the age-related changes in GWFC as well. While GWFC does show a modest increasing trend with age, the spatial extent of these changes is more limited, which aligns with our expectations. These results are now included in the supplementary material for completeness.

Significant Age-Related Variation in GWBPR and GWFC.

- (a) Correlation Between Power Ratio and Age in the HCP-D Cohort: A spatial map (ROI-wise) of the correlation between power ratio and age is presented, with warmer colors (red-yellow) of each ROI indicating stronger positive correlations.
- (b) Correlation Between GWFC and Age in the HCP-D Cohort.

2. Causal language, e.g., on page 4 “High values of GWFC indicate the BOLD signal in WM closely follows the time course of neighboring GM.” Another example on page 5: “Supporting our hypothesis that GWFC is more involved in the transmission of signals to distant regions away from GM”

• Could it be that the signal of origin is from GM at a distance away (ie via longer range connections), and

it is the neighbouring GM that is closely following the WM signal? I also interpret this phrasing as implying that GM signal is more important than WM signal. This may be fine but if it is not the authors' intention, they may wish to rephrase.

Thank you for pointing out these issues. We have carefully reviewed the manuscript and identified several instances of causal language. We have revised them appropriately, as detailed below.

High values of GWFC indicate the BOLD signal in WM closely follows the time course of neighboring GM. -> High values of GWFC reflect strong temporal similarity between BOLD signals in neighboring GM and WM regions.

By comparing GWFC to ReHo, we aim to demonstrate that GWFC provides unique insights into signal changes as it transitions from GM to WM. -> By comparing GWFC to ReHo, we aim to demonstrate that GWFC provides unique insights into functional coupling across the GM-WM boundary.

Again, plausibly, higher-order regions, which exchange more integrated and complex information, engage higher metabolic demands and greater BOLD signal power to send signals away from GM.-> Again, plausibly, higher-order regions, which exchange more integrated and complex information, engage higher metabolic demands and enhanced neurovascular coupling across the GM-WM boundary.

Additionally, long-range functional connectivity (the mean FC between a GM point and all WM points across the brain) is better explained by GWFC than by ReHo, supporting our hypothesis that GWFC is more involved in the transmission of signals to distant regions. -> Additionally, long-range functional connectivity (the mean FC between a GM point and all WM points across the brain) is better explained by GWFC than by ReHo, supporting our hypothesis that GWFC is more involved in the information exchange between distant regions.

3. Use/interpretation of ReHo results

- In the comparison between ReHo and GWFC, two variables are changing: The analysis method and the voxels to which the method is being applied. As a result, the interpretation that GWFC provides “unique insights” may be too strong. It could be that the unique insights are related to the use of Pearson's rather than KCC. Did the authors consider a more apples-to-apples comparison such that Pearson's correlation is applied neighbouring voxels tangential to the boundary, in order to evaluate whether the GWFC method is providing unique insights?

KCC is particularly advantageous for ReHo because it quantifies the consistency of temporal dynamics across a group of voxels, rather than just pairwise relationships. This makes it well-suited for capturing local functional synchrony in a robust and non-parametric manner. However, to address your question directly, we performed a test using a subset of HCP subjects (the first 100 subjects sorted by study ID), replacing KCC with the mean of all pairwise Pearson correlation coefficients among neighboring voxels, and compared the results. As shown in the figure below, the two ReHo measures are highly correlated ($r = 0.9991$), indicating strong agreement between the methods. Importantly, even with this alternative calculation, ReHo still shows a weaker correlation with long-range functional connectivity ($r = 0.31$) compared to GWFC ($r = 0.67$, as shown in Figure 3). These findings suggest that the method of ReHo calculation does not affect the main conclusions of the manuscript. We have added these results to the supplement file.

Comparison of ReHo Calculated Using KCC and Pearson's Correlation. The left panel shows the spatial distribution of ReHo computed using KCC, along with its corresponding relationship with long-range functional connectivity (displayed below). The right panel presents the distribution of ReHo computed using Pearson's correlation and its associated relationship with long-range functional connectivity (also shown below). The spatial correlation between the two measurements is 0.9991.

4. The finding that GWFC is negatively correlated with the sensorimotor-association axis but positively correlated with long range FC seems on its surface to be somewhat contradictory to me. Association areas are involved in very distributed brain networks and have many long range connections. What might be the mechanism for or explanation of this apparent contradiction?

We thank the reviewer for this insightful question. While the observed negative correlation between GWFC and the sensorimotor-association axis, alongside a positive correlation with long-range functional connectivity (FC), may seem contradictory at first glance, this pattern can be explained by the distinct structural and functional profiles of sensorimotor versus association regions.

First, we would like to clarify our use of the term "long-range." In this context, it refers to connections that extend beyond the gray-white matter boundary. The term "long" is relative to GWFC, which captures FC across only a few millimeters.

GWFC reflects the local synchrony of BOLD signals across the GM-WM boundary and is influenced by both structural organization and signal characteristics in these regions. Sensorimotor regions, where GWFC is highest, are characterized by highly myelinated and coherently organized projection fibers, which cross the boundary perpendicularly with minimal crossing or branching. These regions tend to engage in fast, precise, and unimodal processing, resulting in simpler, more uniform signals that are less mixed and more tightly coupled between GM and WM. This structural and functional simplicity supports high signal fidelity, leading to stronger GWFC and more robust long-range FC.

In contrast, association areas support integrative, higher-order cognitive functions and are structurally and functionally more complex. They are connected via heterogeneous and crossing fiber systems (e.g., U-fibers and long-range association fibers), reducing the coherence of WM signal pathways. Functionally, they process diverse and converging information streams, leading to more mixed or multiplexed signals in GM regions and WM regions they connect to, which may reduce the temporal synchrony, regardless of whether the connections span short or long distances.

Minor points:

1. I think the title is misleading. I was expecting the manuscript to be about differences in functional contrast across the boundary. Why not use the “connectivity” as the method is described throughout the manuscript?

We appreciate the reviewer’s comment and the opportunity to clarify our intention. While the term “connectivity” indeed describes the methodological framework of our analysis, we chose the term “functional contrast” deliberately to emphasize a broader concept: the distinct functional characteristics observed at and across the gray-white matter boundary, as reflected by two complementary measures, GWFC and GWBPR. These measures are not limited to connectivity but capture different aspects of signal coordination and power distribution across tissue types, which together form a type of functional contrast. Our use of the term parallels how “contrast” is used in structural imaging (e.g., T1-weighted GM-WM contrast) to describe meaningful signal differences at the boundary. Therefore, we respectfully believe that the current title accurately reflects the conceptual scope of the manuscript, which goes beyond traditional connectivity analysis to highlight functionally distinct profiles across the GM-WM interface.

2. “In addition, due to its structural complexity, the boundary is particularly vulnerable to various diseases and injuries, such as abscesses⁵, metastasis⁶, and diffusive axonal injury⁷”

• What is meant by “structural complexity”? Is the boundary more complex than gray matter itself?

Thank you for the thoughtful question. By “structural complexity,” we are referring to the intricate microstructural organization of the gray-white matter boundary, which is characterized by sharp transitions in cell types, myelination, and axonal orientation. These overlapping structures contribute to local heterogeneity, making the boundary structurally intricate, though not necessarily more complex than gray matter itself as a whole. We have revised the manuscript to clarify this point as below:

“ In addition, due to its intricate microstructural organization, the boundary is particularly vulnerable to various diseases and injuries...”

3. “Despite its importance, the functional interaction across the boundary, which potentially measures signal alterations that occur when it traverses from one type of tissue to another, remains poorly understood.”

• Please revise this sentence. “functional interaction” is not a measurement and it is unclear what “it” is referring to.

Thank you for pointing this out. We agree that the original sentence could be made clearer, particularly with regard to the use of “functional interaction” and the ambiguity of “it.” We have revised the sentence for clarity as follows:

“Despite its importance, the nature of functional coupling across the boundary, which potentially reflects alterations in BOLD signals as they traverse from one type of tissue to another, remains poorly understood.”

4. Figure 2 & 3: how were the scale bar ranges selected for the different measures?

In Figures 2 and 3, the color bars for GWFC and GWBPR were chosen based on the value ranges observed in their respective scatter plots. For example, in Figure 2, the scatter plot showing the

relationship between GWFC and myelination indicates that GWFC values range from approximately 0.25 to 0.6. Accordingly, we set the color bar range from 0.2 to 0.7 to safely encompass all possible values. However, it is important to note that, for myelination, which ranges from about 1.2 to 2.4, we selected a slightly lower minimum value in the color bar to ensure visual consistency with the other two measurements. Nevertheless, the selected range still covers all values shown in the scatter plot.

5. Page 7: “Correlation analyses confirm that the myelin map is more strongly associated with GWFC ($r = 0.40$, $p = 2.7470e-15$) than by ReHo ($r = 0.32$, $p = 2.9541e-10$) Is this a meaningful/significant difference in correlation coefficients? How was this evaluated?”

We thank the reviewer for raising this important point. We performed a Steiger’s Z test to statistically compare the correlation between GWFC and myelin ($r = 0.40$) versus ReHo and myelin ($r = 0.32$), based on 360 cortical regions. The result ($z = 1.726$, $p = 0.086$) indicates that the difference is not statistically significant at conventional thresholds.

However, to further assess the robustness and reproducibility of this pattern, we conducted an internal validation analysis. Specifically, we randomly selected 100 subjects from the full dataset ($n = 687$) and repeated the group-level GWFC, ReHo, and Myelin computations across the cortical surface. We then computed the correlation of GWFC/ ReHo with the Myelin map across regions. This process was repeated 30 times. Across these 30 random subsets, GWFC consistently showed stronger correlations with myelin than ReHo, suggesting that the observed trend is stable and reproducible. We have added this result to the supplement file.

Figure Sx. Comparison of correlation strength between GWFC and ReHo with myelin across 30 random subsets of subjects. To assess the robustness of the relationship between functional metrics and myelin content, we randomly selected 100 subjects (without replacement) from the full cohort ($n = 687$) and computed group-level GWFC and ReHo maps for each subset. For each iteration, we calculated the spatial correlation (r) between each functional metric and the cortical myelin map (across 360 regions). Boxplots summarize the distribution of r values across 30 iterations.

In addition, we note that GWFC consistently shows a stronger and more functionally interpretable relationship with both myelin content and long-range functional connectivity compared to ReHo. This supports our conceptual distinction: GWFC captures signal synchrony perpendicular to the GM-WM boundary, likely reflecting signal fidelity along myelinated projection fibers, while ReHo reflects tangential

synchrony within GM. We believe the observed differences between GWFC and ReHo remain meaningful in terms of biological interpretation and highlight the complementary nature of these two metrics.

We have carefully reviewed the manuscript to make sure we only claim: "Myelin can be better explained by GWFC", instead of "GWFC explains significantly more variance."

6. Why are statistical corrections for multiple comparisons only introduced on page 10? Is a Bonferroni correction appropriate? Why are the other stats from previous pages not corrected for multiple comparisons?

We thank the reviewer for this important question. Statistical corrections for multiple comparisons were introduced on page 10 in the context of neurotransmitter receptor analyses because that section involved testing correlations across 48 receptor maps, a classic multiple comparison scenario. In this case, we applied a Bonferroni correction as a conservative approach to control the family-wise error rate. In earlier analyses (e.g., correlations between GWFC/ReHo and myelin, long-range FC, or the sensorimotor-association axis), we reported a small number of pre-specified correlations, each addressing distinct and hypothesis-driven relationships. Given the low number of comparisons and the strong effect sizes (e.g., p-values < 1e-10), we did not apply multiple comparisons correction at that stage.

7. Page 12 refers to Figure 8; should be Figure 7

We apologize for this error. It has been corrected.

8. Page 20: Should be KCC not KKC

We apologize for this error. It has been corrected.

Reviewer #2 (Remarks to the Author):

The purpose of the study is to introduce two novel fMRI-derived metrics, gray-white matter functional connectivity (GWFC) and gray-white BOLD power ratio (GWBPR). These novel metrics are meant to quantify functional relationships at the gray-white matter boundary in the brain.

Additionally, these novel metrics are compared to other common metrics, such as myelin levels and GM regional homogeneity (ReHo). Lastly, a cohort of 8-21-year-olds were used to determine the trajectory of GWFC and GWBPR during maturation.

This is an important and unique study that should be published once the major and minor points below are addressed. The functional measures of the WM-GM boundary may be an important region to identify the timeline for disease/injury or for the timeline of learning, but clarity is needed to better convey the importance of these novel metrics and the limitations of them.

We sincerely thank you for your positive evaluation of our work.

Major points:

1. These novel functional measures of the WM-GM boundary may be an important region to identify the timeline for disease/injury or for the timeline of learning. It would be especially interesting to longitudinally study these novel metrics along with structural measures either during disease or learning. It would be interesting to see the progression of GWFC with cognitive aging/dementia or demyelinated diseases to see if this boundary is affected before the rest of the myelinated axon. This is one example of interesting future work that can stem from this, however, the future work section and meaning/future implications of the adoption of these measures in the discussion is rather scarce.

We thank the reviewer for this thoughtful and forward-looking comment. We fully agree that one of the most promising directions for future work is the application of our boundary-based functional metrics, such as GWFC and GWBPR, in longitudinal studies of learning, development, and disease progression. As the GM-WM boundary reflects a structurally complex and functionally active interface, we anticipate that it may serve as an early indicator of microstructural or functional disruption, particularly in conditions involving demyelination, axonal injury, or neurodegeneration. In this context, tracking GWFC and GWBPR over time could offer insights into whether the boundary exhibits early alterations preceding widespread white matter changes or whether it reflects adaptive plasticity in learning contexts. We have now expanded the Discussion section to highlight several potential applications of these metrics as below:

“Our findings also suggest promising avenues for future research using GWFC and GWBPR as novel markers of structure-function coupling at the gray-white matter boundary. Given the unique microstructural properties of the boundary, where diverse axonal projections converge and myelination gradients shift, these functional measures may be sensitive to early changes in development, learning, and neuropathology. For example, longitudinal studies could explore whether GWFC is altered in the early stages of demyelinating diseases (e.g., multiple sclerosis) or neurodegenerative conditions (e.g., Alzheimer’s disease), potentially serving as an early functional indicator before widespread white matter disruption. Similarly, boundary metrics may provide insight into experience-dependent plasticity during learning or cognitive training, particularly in association cortices where myelination is prolonged into adulthood. Future work integrating these boundary measures with structural imaging across time could help clarify whether functional changes at the GM-WM interface precede, accompany, or follow underlying

anatomical changes.”

2. The main text of the manuscript (not including the methods) does not mention the demographics of the data used. There is only one small mention of “(8-21)” in the introduction when mentioning age findings. Please mention that HCP datasets are used, which ones, and that this data comes from 8-21 and 22-35 year-olds. HCP-D is mentioned in the results but without any context of the datasets, the reader does not even know that openly sourced data is used for the current study. Along with this, it should be stated that the myelin maps and the neurotransmitter maps are both from different datasets and are different from the fMRI datasets. This should be explicitly stated and mentioned as a limitation.

We thank the reviewer for pointing out this important omission. In the revised manuscript, we now clearly state in the main text (the first paragraph in the Results section) that our study utilized publicly available data from the Human Connectome Project (HCP), specifically the HCP-Development (HCP-D; ages 8–21 years) and HCP-Young Adult (HCP-Y; ages 22–35 years) datasets. These datasets were used for all fMRI-based analyses, including the computation of GWFC and GWBPR.

We have also clarified in the Results section (immediately before the myelin content is introduced) that the myelin maps (based on T1w/T2w ratios) were derived from the HCP-Y structural dataset.

In addition, we have clarified that neurotransmitter receptor maps were derived from separate datasets independent of the HCP imaging data. These externally sourced maps were used to examine spatial associations with our boundary-based metrics, and we now explicitly state in the Discussion that this cross-dataset integration is a limitation.

3. Figure 5 mentions attribute under each of the 6 neurotransmitter receptors that I’m assuming refer to factors the neurotransmitter receptor is associated with [e.g. Serotonin receptor 1F includes Pain (Migraine) modulation]? This is never explained. Also, please state where these associations were pulled from.

We thank the reviewer for pointing this out. The functional labels under each receptor in Figure 5 (e.g., “Pain (Migraine) Modulation” for serotonin 1F) are meant to represent well-established associations from prior literature. These labels are not experimental findings from our dataset but rather serve as interpretive annotations to aid the reader’s understanding of the receptor’s general role in the brain or body. To clarify, we have now added a sentence in the figure legend explicitly explaining this, and we have provided references for each functional association where those associations are mentioned.

Minor points:

1. The abstract does not convey the literature gap (specifically white matter fMRI understanding) that is brought up in the introduction.

We thank the reviewer for this helpful observation. We agree that the original abstract did not sufficiently articulate the broader literature gap regarding the limited understanding of white matter fMRI and the functional role of the gray-white matter boundary. In response, we have revised the abstract to explicitly state this gap and better connect our motivation to the broader context of the field. The updated abstract now highlights the underrepresentation of WM in traditional fMRI research and frames our proposed boundary-based metrics (GWFC and GWBPR) as tools to address this overlooked aspect of brain function. We believe this revision improves the clarity and impact of the contribution of our study.

2. Throughout the paper, GWFC is referred to as GWFC and correlation and connectivity. Similarly, GWBPR is sometimes referred to as power ratio. Please be consistent with using the defined abbreviations instead of introducing different phrases for these measures.

We appreciate the reviewer's careful reading. In response, we have thoroughly revised the manuscript to ensure consistent use of the defined abbreviations. Specifically, we now refer to "GWBPR" throughout the text in place of "power ratio."

3. The introduction mentions that these novel functional metrics complement traditional structural assessments of the GM-WM boundary. Are you just referring to the structural contrast of T1w imaging? If there is more that you're referring to, please expand on this.

We thank the reviewer for this important question. In the Introduction, our statement that the proposed functional metrics complement traditional structural assessments of the GM-WM boundary specifically refers to the T1-weighted intensity ratio between adjacent GM and WM points (structural contrast), measured along directions perpendicular to the boundary. To clarify our intent, we have revised the text to emphasize that our functional metrics are meant to complement structural contrast measures, rather than general structural assessments, at the GM-WM interface. This aligns more precisely with the theme of the manuscript and our title, which focuses on "functional contrast."

4. As there are many different methods to quantify myelin, in the introduction, it may be important to briefly mention that the current study uses the T1w/T2w ratio method.

We thank the reviewer for this helpful suggestion. We agree that it is important to clarify the specific method used to quantify myelin content. In response, we have revised the Introduction to briefly state that the current study uses the T1-weighted to T2-weighted (T1w/T2w) ratio to estimate cortical myelin content.

5. The sensorimotor-association map should be briefly explained in the introduction or results or figure caption to provide a better idea of what the map is telling us. This would help highlight the importance of GWFC association with it and the discussion points linking high GWFC with more basic sensorimotor functions, without having to dig deep into the methods to understand what the map was.

We thank the reviewer for this insightful suggestion. We have added a brief introduction of the axis immediately after it is mentioned in the results section.

6. The third-last and second-last paragraphs in the introduction discuss the results and the potential implications of them. These paragraphs and discussions belong in the discussion section, not the introduction.

We have removed all discussion-like statements from these two paragraphs and confirmed that the deleted content is already fully addressed in the Discussion section.

7. Figures 2, 3, 4, and 6 should include letters to refer to in the caption for organization. Similar to what you have done in other figures.

We thank the reviewer for this helpful suggestion. We have added panel labels to Figures 2, 3, 4, and 6 to improve organization, consistent with the formatting used in our other figures. Corresponding references to these panels have also been added in the figure captions and main text where appropriate.

8. The figure 2 schematic should be better explained. I.e. state that the top and bottom schematics refer to high and low GWFC/ReHo.

We thank the reviewer for pointing this out. These diagrams are intended to conceptually illustrate how different orientations of functional connectivity, perpendicular for GWFC and tangential for ReHo, may reflect varying underlying anatomical and myelin characteristics. Your understanding is correct. The top and bottom schematics, aligned with the color bars, correspond to regions with high and low GWFC or ReHo values, respectively. The revised caption provides this clarification for improved interpretability.

9. This is a limitation that should be mentioned in the main text and the potential meaning/result/implication of this limitation in voxels and their size should be mentioned: “The WM point lies along the normal line extending from v_i to WM, positioned 1 mm away from the boundary surface. Note that the fMRI voxel size in this study is 2 mm isotropic and thereby trilinear interpolation was applied to estimate the BOLD signal at sub-voxel precision at those points and vertex. It is important to note that, as some subcortical structures are physically close to certain boundary areas, such as putamen, the interpolation process could introduce GM signals into the measured signal of WM points. To eliminate this, we set the fMRI intensity of all subcortical structures to zero before calculating GWFC and the power ratio.” Could also mention the limitation of the myelin measure being an average of all cortical layers instead of just the value at the vertex.

We thank the reviewer for this important observation. We have now stated this limitation in the first paragraph of the results section and clarified the potential implications, namely, that local signal mixing could influence the accuracy of WM signal estimation. We also stated it in the methods section. This limitation may be further addressed in future studies by using higher-resolution fMRI or alternative sampling strategies less sensitive to partial volume effects.

To address your second question directly, we compared the myelin maps calculated using the average across all cortical layers with those calculated on the mid-thickness surface. The latter was computed using a subset of HCP subjects (the first 100 subjects sorted by study ID). The strong correlation between the two estimates supports the robustness of our findings despite the use of averaged cortical layer values. This result has been added to the supplement file.

Comparison of myelin map calculated based on the average of all cortical layers and on the mid-thickness surface. Note that the later are computed based on a subset of HCP-Y subjects (the first 100 subjects sorted by study ID).

10. The orientation of intersecting WM fibers is a limitation of the long-range correlation of GWFC and its correlation with WM microstructure, correct? Specifically based on your discussion: “The same diffusion MRI study²⁷ revealed that within superficial white matter regions containing U-fibers and association

fibers, the orientation of fibers tends to be more complex and variable. In these regions intersecting signals²⁸ potentially mix or cancel each other out, reducing their correlation with cortical activity. In contrast, projection fibers are characterized by more consistent and coherent orientations, often aligning perpendicular to the GM-WM boundary, supporting more synchronized signal transmission across the boundary.”

We thank the reviewer for this thoughtful point. We agree that complex or crossing fiber orientations, particularly in regions dominated by U-fibers and long-range association fibers, may reduce the spatial coherence of the BOLD signal in white matter, thereby weakening the measured GWFC. This could, in turn, attenuate the observed correlation between GWFC and long-range FC in these regions. However, we view this as a measurement limitation rather than a conceptual contradiction. Our original interpretation that higher GWFC reflects more effective signal transmission from GM into WM and is, therefore, more predictive of long-range FC remains valid, particularly in projection-dominated regions, where fibers are more coherently aligned perpendicular to the boundary.

Reviewer #2 (Remarks on code availability):

The code could include more comments to make it easier to understand input/output data types and sizes.

We appreciate the reviewer's suggestion. We have revised the code to include additional comments that clarify input and output data types.

Reviewer #3 (Remarks to the Author):

Reviewer #3 (Remarks on code availability):

The ReadMe does not include installation or use instructions. There could be more comments in the code to make it easier to understand and to better understand input/output data types and sizes. There are some comments to get a brief overview of what some code sections are doing.

We appreciate the reviewer's suggestion. We have revised the code to include additional comments that clarify input and output data types. We also include other instructions in the ReadMe file.

Reviewer #4 (Remarks to the Author):

This manuscript introduces two new measures of BOLD functional connectivity: gray-white matter FC (GWFC) defined as the BOLD correlation between each pair of points perpendicular to the GW-WM boundary along the brain surface, and the gray-white BOLD power ratio (GWBPR) defined as the ratio of BOLD fALFF between the two points (GM vs WM) of each pair. The authors go on to compare the resulting brain surface patterns for GWFC and GWBPR and also for ReHo with surface patterns for T1w/T2w ratio (a proxy for myelination), neurotransmitter density distributions and sensori-motor association axis. They also examine the evolution of GWBPR with brain development during adolescence.

The ideas are interesting, and the manuscript is written clearly. My utmost concern is the partial volume effects between cortex and subcortical WM, that the authors describe as a limitation but take no specific precautions or sanity checks to convince the reader that this PVE does not drive the results of the paper entirely.

We sincerely thank you for your thoughtful evaluation of our work. As your main concern is elaborated in Major Point 1, we have addressed it directly in our response to that section.

Major points:

1. Specifically, the point in WM perpendicular to the surface is taken 1mm from the WM-GM boundary, for an fMRI acquisition with 2-mm resolution. How was this 1-mm distance chosen? It is crucial that the authors test any potential spatial pattern association of their metrics with cortical thickness and gyrus/sulcus curvature, to ensure the associations are not driven by local brain morphology that would introduce more or less BOLD PVE of GM into WM.

We thank the reviewer for this important question. The choice to sample the WM signal 1 mm beneath the GM-WM boundary was guided by prior work examining tissue intensity contrast using T1-weighted MRI. Specifically, Salat et al. (2009, NeuroImage) sampled white matter at 1 mm below the boundary using T1-weighted MRI at ~1.3 mm voxel resolution. Despite the voxel size being larger than 1 mm, this distance was chosen as a conservative and practical offset to target superficial white matter. This 1 mm offset has become a standard and conservative practice in surface-based studies of gray-white contrast, ensuring consistency with prior neuroimaging methodologies.

We acknowledge that the resolution of our fMRI data (2 mm isotropic) is notably lower than that of the T1-weighted data used in Salat et al. (2009), which may introduce more pronounced partial volume effects from GM into WM. An important consideration in addressing this concern is that GWFC measures the synchronization of BOLD signals across the GM-WM boundary. If partial volume effects were a dominant factor, regions with greater contamination from GM into WM would likely artificially increase GWFC values by enhancing signal similarity between adjacent GM and WM voxels. On the other hand, GWBPR measures the power ratio between WM and GM, which is typically less than 1 due to lower vascularization in WM. Since GWBPR also reflects a form of signal similarity across the boundary, higher partial volume effects would be expected to increase GWBPR as well (i.e., shift it closer to 1) in the same regions where GWFC is elevated. However, our results show that GWFC and GWBPR exhibit largely

inverse spatial distributions. This divergence suggests that partial volume effects are unlikely to be the primary driver of the observed patterns.

To directly address concerns, we conducted a control analysis by sampling white matter signals at a 4 mm offset, equivalent to two fMRI voxels, beneath the GM-WM boundary. At this depth, any contamination from the gray matter signal is expected to be minimal. As shown in the figures below, the spatial distributions of both GWFC and GWBPR (based on 200 subjects) remained consistent with our original findings, and their correlations with the myelin content were preserved. These results support the robustness of our conclusions and suggest they are not driven by contamination from GM. This clarification has been added to the revised manuscript, and the results of the 4 mm control analysis are included in the supplementary files.

Comparison of GWFC maps calculated using WM voxels sampled at different distances from the GM-WM boundary.
(a) Spatial distribution of GWFC calculated using WM voxels located 1 mm beneath the boundary.
(b) Spatial distribution of GWFC calculated using WM voxels located 4 mm beneath the boundary.
(c) Relationship between GWFC (4 mm) and cortical myelin content, demonstrating that the spatial correlation between GWFC and myelin is preserved even when WM voxels are sampled farther from the boundary, indicating robustness against partial volume effects.

Comparison of GWBPR maps calculated using WM voxels sampled at different distances from the GM-WM boundary.
(a) Spatial distribution of GWBPR calculated using WM voxels located 1 mm beneath the boundary.
(b) Spatial distribution of GWBPR calculated using WM voxels located 4 mm beneath the boundary.
(c) Relationship between GWBPR (4 mm) and cortical myelin content, demonstrating that the spatial correlation between GWBPR and myelin is preserved even when WM voxels are sampled farther from the boundary, indicating robustness against partial volume effects.

Moreover, we conducted a test for long-range FC by comparing results based on the original WM mask and a version eroded by 4 mm. As shown below, the two distributions are highly correlated ($r = 0.9974$), and the distribution based on the eroded mask still captures the sensorimotor–association variation. This supports the robustness of our findings.

Other major revision points that would go in the direction of reinforcing confidence in the message of the manuscript are listed below. Further minor points follow.

2. The authors seem to imply that there is a consistent directionality of information transfer from GM to WM, e.g. “how much of the power of the GM BOLD signal is retained as it traverses the boundary”. However, this directionality varies: sometimes the WM connections are efferent into the GM, e.g. in the case of the visual system, the information comes from the optic nerves into the LGN and is then relayed to the visual cortex via the optic radiations. How does this *actual* directionality impact the analysis and the interpretation? The situation for higher order functions is even more complex in terms of information flowing into or out of the GM and from or into the WM.

We thank the reviewer for this important observation. We fully agree that in the brain, signal flow across the GM WM boundary is bidirectional and regionally variable depending on the functional system, task demands, and anatomical organization. Our study does not attempt to infer the direction of information flows across the boundary. Instead, GWFC and GWBPR are designed to quantify the degree of coupling or relative signal power between adjacent GM and WM tissue. While we use language such as “signal traversing the boundary” to describe these relationships, this is not intended to imply a fixed or unidirectional flow from GM to WM. We have revised relevant parts of the manuscript to clarify that our interpretations are functionally descriptive and not meant to imply strict anatomical causality. Specifically, we revised all causal languages appropriately and replaced the “retention” with “alteration”. Thank you for prompting this clarification.

3. It seems to me that long-range connectivity resembles in fact some sort of seed-based analysis, looking at the connections of the seed in the WM. Why not show such connectivity spatial maps for

example seeds in the GM? Also, does it make sense to average over the whole WM for each seed, rather than over voxels significantly correlated with the seed in the WM?

We thank the reviewer for this thoughtful observation. Indeed, our long-range FC measure shares some similarities with seed-based analysis in that it involves computing the correlation between a seed region (each GM vertex) and a set of voxels (in this case, across the WM). However, our intention was not to generate spatial FC maps from each seed but rather to derive a scalar summary measure representing the overall degree of functional coupling between a GM location and the white matter as a whole. We chose to average over all WM voxels, rather than only significantly correlated voxels, in order to produce a consistent and unbiased measure that could be compared across the cortex. This approach avoids setting arbitrary thresholds and ensures that the resulting long-range FC values are not confounded by differences in SNR, local variability, or spatial extent of correlations.

That said, we agree that visualizing full spatial correlation maps from selected GM seed points can be informative. In response to this suggestion, we have generated long-range FC maps for two representative GM clusters, specifically, two regions that exhibited the highest GWFC values. As shown in the figures below, the resulting voxel-wise FC maps illustrate the distributed coupling patterns between each GM cluster and white matter voxels across the brain. These maps confirm that the long-range FC values used in our analyses reflect, to some extent, spatially coherent and anatomically plausible connectivity patterns consistent with known projection pathways from sensorimotor and visual cortices. Specifically, the cluster in the somatosensory region shows strong correlations with deep and distant WM voxels along known corticospinal tracts, while the visual cluster shows widespread correlations with WM regions, particularly within tracts extending toward the posterior of the brain. These patterns suggest that GWFC captures functionally relevant signal exchange across the GM-WM boundary. We have included these results in the supplementary materials.

Long-range functional connectivity (FC) maps for two representative GM seed clusters with high GWFC values.

- (a) Surface rendering of the first GM cluster.
- (b) Surface rendering of the second GM cluster.
- (c) Voxel-wise FC map showing correlations between all WM voxels and the GM cluster shown in (a).
- (d) Voxel-wise FC map showing correlations between all WM voxels and the GM cluster shown in (b).
- (e) Atlas-based distribution of corticospinal tracts.
- (f) Atlas-based distribution of posterior thalamic radiations.

4. Related to this point, if the assumption is that myelin transmission is fast and metabolically non-demanding, then the GWFC should be preserved across WM voxels belonging to the same fiber to which the initial WM voxel considered close to the boundary belongs. Showing high GWFC along a whole bundle would increase the confidence that this metric is not heavily affected by PVE in proximity to the cortex.

We thank the reviewer for this thoughtful point. In our response to your question 3, we generated voxel-wise long-range FC maps from two representative GM clusters that exhibited the highest GWFC values. These maps show widespread WM correlations extending well beyond the boundary, to some degree

consistent with known anatomical pathways. Together with our control analysis (in response to your question 1) using WM points sampled 4 mm beneath the boundary, these findings suggest that GWFC is not merely a boundary artifact caused by partial volume effects but reflects coherent functional organization along white matter tracts.

5. Too many statements are made using “plausibly”, without literature references to back them up. If there is no clear literature reference supporting these statements, please say so explicitly. This impression of speculative statements is reinforced by the fact that the Introduction already develops substantially on all the findings and interpretations of the paper, which better belong in Discussion alone.

We thank the reviewer for this valuable suggestion. First, we have removed all discussion-like statements from the Introduction and confirmed that the deleted content is fully addressed in the Discussion section. We also carefully reviewed the manuscript to identify speculative statements. Where supporting references were available, we have cited them appropriately. For statements lacking direct empirical support, we have explicitly stated that they only represent our interpretation of the findings.

6. Why would higher order regions require greater BOLD signal power (i.e. higher ratio of fALFF in the WM point vs fALFF in the matching GM point across the border) ? Is the WM there more vascularized? Do they involve more subcortical regions that act as relays, which are often described as a mixture of GM and WM, with both vascularization and myelin?

We appreciate the reviewer’s question and the opportunity to clarify this point. We believe part of the confusion may stem from how we originally described the interpretation of GWBPR in the manuscript.

To clarify, GWBPR is not an absolute measure of BOLD signal amplitude in WM alone but a relative measure between a WM point and its paired GM point across the GM-WM boundary. It reflects the gradient or contrast in signal power (or fALFF, an approximate indicator of local energy demand or metabolic activity) between the two tissue types at each location. It is known that WM is typically less vascularized than GM throughout the brain, and thereby, the GWBPR values are always less than 1. Our findings show that lower GWBPR values are typically observed in lower-order regions, which support relatively simple, rapid signal transmission in WM. In these regions, the WM side may require less BOLD power to propagate information efficiently, resulting in a sharper transition across the boundary. In contrast, higher-order regions are functionally more complex and involved in multimodal integration. The WM side of these regions may require greater metabolic resources to support the transmission of more diverse and complex signals. This results in a shallower gradient between GM and WM power and, thus, higher GWBPR.

It is well established that energy demand is generally supported by higher vascularization. GWBPR reflects a relative contrast in BOLD power across the GM-WM boundary, which may indirectly relate to a gradient in vascularization or neurovascular coupling between the two tissues. That said, we do not have direct evidence regarding regional differences in vascular density across the boundary, and we acknowledge this as a limitation.

Finally, GWBPR provides complementary information to GWFC. While GWFC captures the temporal synchrony between GM and WM signals, GWBPR reflects how different the two sides are in terms of energetic activity, independent of the exact signal pattern. Together, these metrics offer a more comprehensive view of functional signal propagation across the GM-WM boundary. We have revised the Discussion section to clarify this interpretation and thank the reviewer for prompting this important distinction.

7. “thereby require a high degree of plasticity, which however may be impeded by myelination”. Is there a reference for this? Myelination is actually an active component of brain plasticity.

We thank the reviewer for this important point. While there is evidence suggesting that myelination can limit certain forms of neuronal plasticity, for example:

Xin, W., Kaneko, M., Roth, R.H. et al. Oligodendrocytes and myelin limit neuronal plasticity in visual cortex. Nature 633, 856–863 (2024). <https://doi.org/10.1038/s41586-024-07853-8>

We also acknowledge that other studies have highlighted myelin as an active contributor to plasticity. Given these differing perspectives in the literature, we have removed this interpretation from the manuscript to avoid overstating a potentially contentious claim.

8. The relationship between long range FC and GWBPR could also be reported, to complete the analysis of its association with GWFC and ReHo

Thank you for this suggestion. As shown below, we have compared long-range FC and GWBPR and the figure has been added to the supplement files.

Comparison of GWBPR versus long-range FC.

9. Same suggestion for the relationship with neurotransmitter receptor densities. Why were they correlated to GWFC but not other metrics introduced in the paper? E.g. receptors that show negative correlations with GWFC: do they show positive correlation with the BOLD power ratio instead, as of their association with higher order functions?

Thank you for this suggestion. We have calculated the correlations between GWBPR and the neurotransmitter receptors that showed the strongest correlations with GWFC. As shown in the figure below, they exhibit an inverse pattern relative to Figure 5, as expected. This analysis has been included in the supplementary materials.

Correlations Between GWBPR and Neurotransmitter Receptor Distributions.

Left Panel: The GWBPR distribution is shown across the cortex.

Right panel top row: The spatial maps display three selected neurotransmitter receptors positively correlated with GWBPR.

Right panel bottom row: The spatial maps depict three selected neurotransmitter receptors negatively correlated with GWBPR.

The selected neurotransmitter receptors are the three most positively and three most negatively correlated with GWFC, included for ease of comparison with Figure 5.

10. P. 13-14: § “Although ReHo [...] the presence of even low levels of myelination still plays a role in improving the precision and timing of local neural signals, contributing to the significant correlation.” ◇
 What range of timings and delays do the authors mean? The precision and timing of local neural signals, with or without myelin, is likely much faster than the BOLD response and the temporal resolution of the acquisition (TR = 720 or 800 ms).

We thank the reviewer for pointing this out. We agree that the statement regarding the “precision and timing” of neural signals was misleading in the context of BOLD fMRI, as the temporal resolution of our data (TR = 720–800 ms) is insufficient to capture the fast timescales of neuronal conduction. We have revised the sentence to remove this phrasing regarding timing.

11. P.16: “The dense [...] isolated channels.” Can the authors provide a reference supporting this?

We apologize for the misleading use of the term 'isolated.' What we intended to convey was 'insulated.' We have revised the manuscript accordingly and now use the following two references to support the claim that higher myelination is associated with reduced energy costs and enhanced insulation.

Saab, A. S., Tzvetanova, I. D. & Nave, K.-A. The role of myelin and oligodendrocytes in axonal energy metabolism. *Curr. Opin. Neurobiol.* 23, 1065–1072 (2013).

Myelin insulation as a risk factor for axonal degeneration in autoimmune demyelinating disease. *Nat. Neurosci.* 26, 1218–1228 (2023).

Minor points:

1. The reference to the GM-WM boundary and the information “exchanged” between tissue types is a bit surprising to me. Is this description backed by literature? A simpler picture would be neuronal cell bodies in the gray matter projecting axons into the WM to create synapses with distant areas, but why this notion of information exchange at the boundary?

We thank the reviewer for raising this important point. Our use of the term “information exchange” at the GM-WM boundary is intended to conceptually describe the interface where neural signals transition from predominantly local processing in gray matter to long-range transmission through white matter tracts. We understand the concern and have revised the manuscript as below to clarify that this term is not meant to imply a unique form of exchange at the boundary:

The boundary between GM and WM, linking local cortical processing with broader white matter pathways that support large-scale brain communication, represents a unique and critical interface for understanding brain connectivity.

2. P.12: “adolescent group shows widespread reductions in the power ratio compared to young adults”: better to phrase this as lower power ratio as compared to young adults, since the data is cross-sectional.

We thank the reviewer for this helpful clarification. We agree that “reductions” implies a longitudinal interpretation, which is not appropriate given our cross-sectional design. We have revised the sentence to use “lower power ratio” instead of “reductions” to more accurately describe the group-level difference between adolescents and young adults.

3. It is a pity that subcortical regions were masked out, if I understood correctly. Indeed, they act as relays which would have been interesting to study in the context of this work.

We agree with the reviewer’s concern and recognize that the exclusion of subcortical structures limits the scope of our analysis. This decision was made to minimize partial volume effects and contamination of WM signal estimates due to the spatial proximity of these deep GM regions. However, we acknowledge that subcortical structures play a critical role as relay centers in brain communication, and their exclusion may have omitted important aspects of boundary-related functional dynamics. To address this, we have added the following limitations to the manuscript:

The exclusion of subcortical structures also prevented us from examining the functional role of subcortical structures, such as the thalamus and basal ganglia, which are known to act as key relays in brain communication. Future research using higher-resolution data and improved methods for isolating subcortical contributions may enable a more comprehensive understanding of their role in boundary-related functional dynamics. This issue is a limitation of current imaging techniques, as all surface-based fMRI studies face similar challenges. Nonetheless, future work should aim to utilize higher-resolution data and incorporate subcortical structures into the analysis to provide a more complete picture of large-scale signal integration.

Reviewer #1 (Remarks to the Author):

The authors have thoroughly addressed the concerns raised. I have two minor observations related to the abstract.

Note age range (8-35, ie not older adults)

The abstract defines the sensorimotor-association axis differently (with a focus on receptors) compared to the results which on page 10, line 194 describes the axis as being derived from various techniques.

We sincerely thank you for your positive evaluation of our work.

We have now included the age range for the development group (8–21 years) in the abstract as requested.

We apologize for the confusion regarding the sensorimotor–association axis. GWFC aligns with both the sensorimotor–association axis and receptors associated with basic functions, which are two distinct measurements. In our previous abstract, we only mentioned the receptors, which may have led to a misunderstanding. We have now revised the sentence to:

“GWFC aligns with patterns of myelination, long-range connectivity, and sensorimotor organization, suggesting efficient signal transmission.”

This phrasing describes the general functional relevance of GWFC and implicitly captures the contributions of both the axis and receptor distribution, while staying within the word limit.

Reviewer #4 (Remarks to the Author):

The authors have put substantial effort in the revision, which is appreciated. There are a few important points that would be worth addressing further:

We sincerely thank you for your positive evaluation of our work.

Several reviewers suggested more paralleled analyses of GWFC and GWBPR vs other measures, such as long range FC, ReHo, neurotransmitter densities etc. While the authors have provided all of those in the revised version, it still seems a bit arbitrary which ones go in the main text vs supplementary material. Especially as the correlation of long range FC is stronger with GWBPR (supplementary) than with GWFC (main), and the strengths of correlations with neurotransmitter distributions are very similar (yet GWFC is in main, GWBPR in supplementary). This could suggest the authors made the decision based on what fit their narrative, and while this was most likely not their intention or rationale, it would make the paper stronger if they provided more balanced analyses of the two measures. More importantly, I wonder how strongly anti-correlated GWFC and GWBPR are, and therefore whether they really provide complementary information or are perhaps quite redundant (i.e. one is the inverse of the other).

We thank the reviewer for this insightful comment. Initially, we did not include the GWBPR correlations with other measurements in the main text because we observed a strong negative correlation between GWBPR and GWFC across the cortex. Given that GWFC was already shown to correlate positively with myelination, long-range connectivity, and sensorimotor-related receptors, we anticipated that GWBPR would naturally exhibit inverse correlations with those same features. Therefore, we just included the GWBPR correlations in the supplementary material primarily to confirm this expectation. In the GWBPR section of the main text, we instead focused on highlighting its complementary role to GWFC.

However, we agree that the presentation lacked balance between the two metrics. To address this, we have moved the GWBPR vs. long-range FC analysis to the main text, as it showed a notably stronger correlation than GWFC. Additionally, we now explicitly report the correlation between GWFC and GWBPR ($r = -0.68$) in the Results section. The other comparisons remain in the supplementary material, as their patterns can be reasonably inferred given the strong negative correlation between GWFC and GWBPR.

While GWFC and GWBPR are significantly anti-correlated, they are not completely the inverse of each other, and we believe they offer complementary insights. According to our findings, GWFC appears relatively stable across development, whereas GWBPR shows widespread age-related increases, particularly in higher-order regions that are still maturing during adolescence. This developmental sensitivity of GWBPR supports its unique contribution beyond simply being the inverse of GWFC. Notably, both GWFC and GWBPR in general increase with age, further supporting that they do not behave as strict opposites across all dimensions.

We thank the reviewer again for prompting a more balanced and transparent presentation.

Another point that would be important to understand is the interpretation of low GWFC (so low temporal correlation between GM and WM BOLD timecourses at these boundary points) associated rather with high GWBPR (so high fALFF for BOLD timecourse in WM at that point), and vice versa. The high GWBPR suggests high BOLD power in the expected functional frequency ranges, yet the temporal correlation with the GM BOLD is poor. So could the authors also run cross-correlation of BOLD WM vs

GM timecourses for a range of temporal delays, to check if there is a delay between the two which would explain the low GWFC but high GWBPR? And would accounting for that temporal delay of the BOLD response in WM boost the GWFC for those regions?

We thank the reviewer for this insightful suggestion. To evaluate whether temporal delays between GM and WM BOLD signals could explain instances of low GWFC despite high GWBPR, we computed a new version of GWFC that does not assume zero lag. Specifically, we performed cross-correlation analyses across a range of temporal shifts (−3.6 s to +3.6 s in 0.72 s intervals) between WM and GM time courses at boundary points and extracted the maximum correlation. This analysis was conducted on fifty representative subjects (the first fifty used in our study). Below, we compare the group-averaged “maximum correlation” map (left) to the original zero-lag GWFC map (right). The spatial distributions are nearly identical. We observed a slight increase in FC values in some regions in the maximum correlation maps, indicating that temporal delays do exist and may slightly lower zero-lag correlations. However, regions with originally low GWFC remained relatively low, suggesting that temporal lag is not the primary driver of regional differences in GWFC.

This reviewer had understood the definition of the GWBPR correctly the first time around, there was no confusion involved. My question was whether regions involved in higher order functions had higher GWBPR because of higher vascularization in the WM there, but there seems to be no available information on WM vascularization for different bundles and networks. As a side note, given that the ratio is that of $fALFF(WM)/fALFF(GM)$, I wonder if the measure should not better be called WGBPR than GWBPR.

We sincerely apologize for the misunderstanding. We thank the reviewer again for this thoughtful point. We agree that differences in white matter vascularization could contribute to regional variations in GWBPR, particularly in higher-order regions. As noted in the Discussion, we have acknowledged this as a limitation: “While it is well established that energy demand is generally supported by vascularization, we

do not have direct evidence regarding regional differences in vascular density across the boundary.” We also mention that this relationship may indirectly reflect a gradient in vascularization or neurovascular coupling between GM and WM. We have retained this discussion in the revised manuscript to highlight the need for future work incorporating direct vascular measurements to clarify this interpretation.

We also appreciate the reviewer’s suggestion regarding the naming of the GWBPR metric. We chose GWBPR (gray-white BOLD power ratio) to be consistent with the commonly used term in neuroimaging, where the interface is typically referred to as the gray-white matter (GW) boundary. Additionally, this naming aligns with the other metric, GWFC, and with the established gray-white contrast ratio (GWR), which is computed as $T1(WM)/T1(GM)$ used in structural imaging. To maintain coherence across measures and with existing literature, we have retained the name GWBPR in the revised manuscript.